# Re-Evaluation of the Order Sordariales: Delimitation of Lasiosphaeriaceae s. str., and Introduction of the New Families Diplogelasinosporaceae, Naviculisporaceae, and Schizotheciaceae

**DOI:** 10.3390/microorganisms8091430

**Published:** 2020-09-17

**Authors:** Yasmina Marin-Felix, Andrew N. Miller, José F. Cano-Lira, Josep Guarro, D. García, Marc Stadler, Sabine M. Huhndorf, Alberto M. Stchigel

**Affiliations:** 1Department Microbial Drugs, Helmholtz Centre for Infection Research, Inhoffenstrasse 7, 38124 Braunschweig, Germany; Marc.Stadler@helmholtz-hzi.de; 2Mycology Unit, Medical School and IISPV, Universitat Rovira i Virgili, C/ Sant Llorenç 21, 43201 Reus, Tarragona, Spain; jose.cano@urv.cat (J.F.C.-L.); josep.guarro@urv.cat (J.G.); dania.garcias@urv.cat (D.G.); albertomiguel.stchigel@urv.cat (A.M.S.); 3Illinois Natural History Survey, University of Illinois, 1816 S. Oak St., Champaign, IL 61820, USA; amiller7@illinois.edu; 4Botany Department, The Field Museum, Chicago, IL 60605, USA; shuhndorf@fieldmuseum.org

**Keywords:** *Areotheca*, *Lundqvistomyces*, *Naviculispora*, *Pseudoechria*, *Pseudoschizothecium*, *Rhypophila*, soil, 35 new taxa

## Abstract

The order Sordariales includes the polyphyletic family Lasiosphaeriaceae, which comprises approximately 30 genera characterized by its paraphysate ascomata, asci with apical apparati, and mostly two-celled ascospores, which have a dark apical cell and a hyaline lower cell, frequently ornamented with mucilaginous appendages. To produce a more natural classification of this family, we carried out a phylogenetic analysis based on sequences of the internal transcribed spacer region (ITS), the nuclear rDNA large subunit (LSU), and fragments of ribosomal polymerase II subunit 2 (*rpb2*) and β-tubulin (*tub2*) genes of several isolates from soil and of reference strains of the Sordariales. As a result, Lasiosphaeriaceae s. str. has been circumscribed for the clade including the type species of the genus *Lasiosphaeria* and, consequently, its description emended. In addition, the new families Diplogelasinosporaceae, Naviculisporaceae, and Schizotheciaceae are introduced to accommodate those taxa located far from the Lasiosphaeriaceae s. str. Moreover, we propose the erection of the new genera *Areotheca*, *Lundqvistomyces*, *Naviculispora*, *Pseudoechria*, *Pseudoschizothecium*, and *Rhypophila* based on morphological and sequence data. New combinations for several species of the genera *Cladorrhinum*, *Jugulospora*, *Podospora*, *Schizothecium*, and *Triangularia* are proposed, their descriptions are emended, and dichotomous keys are provided to discriminate among their species.

## 1. Introduction

The order Sordariales [1] is one of the most diverse taxonomic groups within the class Sordariomycetes, and includes taxa characterized by the production of ascomata with membranaceous or coriaceous ascomatal walls and one- or two-celled ascospores often ornamented with appendages or sheaths [2]. Depending on the authors, the order has historically contained 7 to 14 families [1,3], until Huhndorf et al. [2] restricted it to three families, i.e., Chaetomiaceae, Lasiosphaeriaceae, and Sordariaceae based on a phylogenetic study. In that work, Chaetomiaceae and Lasiosphaeriaceae did not form monophyletic clades. Sordariaceae was considered a monophyletic family, but Cai et al. [4] demonstrated that the genus *Diplogelasinospora*, which is still included in the Sordariaceae, was not in the family, being more closely related to Lasiosphaeriaceae.

Several studies have been performed on the Chaetomiaceae to properly delimitate the family and their largest genera, i.e., *Chaetomium* and *Thielavia* [5,6,7]. As a result, the family currently includes more than 35 genera, of which 17 have been recently introduced to accommodate species previously in *Chaetomium* and *Thielavia*, which did not cluster with the type species of these genera.

To date, Lasiosphaeriaceae remains a paraphyletic group. This family, erected in 1932 by Nannfeldt, is the largest and most diverse of the Sordariales [8]. It comprises usually coprophilous plant debris inhabitants and soil-borne taxa that develop paraphysate ascomata with different ascomatal wall structures, cylindrical or clavate unitunicate asci, which usually possess a non-amyloid apical apparatus, and mostly two-celled ascospores with a dark apical cell and a hyaline basal cell, smooth-walled or with different types of ornamentations, with germ pores and usually bearing mucilaginous appendages [9]. In a recent phylogenetic study, the family Podosporaceae was introduced to accommodate the type species of *Podospora* and other lasiosphaeriaceous taxa that grouped in a monophyletic clade far away from the genus *Lasiosphaeria* [6]. However, other lasiosphaeriaceous members are found in other clades located far from the Lasiosphaeriaceae s. str. clade, and their taxonomic placement remains unresolved.

Another problem concerning the Sordariales is that most of the lasiosphaeriaceous genera are polyphyletic [4,6,10,11,12], as the taxonomy of these has historically been based mainly on morphological criteria, which has been demonstrated to be incongruent with molecular phylogenies. Miller and Huhndorf [11] carried out a phylogenetic study including many members of the Sordariales, mostly from the family Lasiosphaeriaceae, and they noted that the morphology of the ascospores was an extremely homoplastic character and could not be used to predict phylogenetic relationships. They also observed that the nature of the ascomatal wall could be used to delimit certain clades. Subsequently, Cai et al. [4] reached the same conclusion, considering the ascomatal wall a better predictor of phylogeny. Based on these criteria, they redefined the genus *Schizothecium* (Lasiosphaeriaceae). Recently, *Apiosordaria*, *Podospora*, and *Triangularia* were studied and delimitated based primarily on DNA sequence data [6], but a large number of polyphyletic genera and species still need to be properly delimited and their taxonomy revised.

In an effort to contribute to the more accurate delimitation of taxa in the Sordariales, especially those that have been assigned to the Lasiosphaeriaceae, we have carried out a phylogenetic study based on the nucleotide sequences of the internal transcribed spacer region (ITS), the large subunit (LSU) of the rDNA, and partial fragments of the DNA-directed RNA polymerase II second largest subunit gene (*rpb2*) and β-tubulin (*tub2*) including a significant set of reference and fresh strains of species isolated from soil belonging to this order.

## 2. Materials and Methods

### 2.1. Soil Sampling and Fungal Isolation

Soil samples were collected in the “Abra del Infiernillo” (Tafí del Valle, Argentina), in Gwalior (India), in the Great Smoky Mountains National Park (an International Biosphere Reserve of USA), and in different locations throughout Spain. For the isolation of soil-borne ascomycetes, we followed a previously described procedure [13] to activate dormant spores by using thermal shock at 60 °C and chemical agents, i.e., 5% *v*/*v* acetic acid and 2% *w*/*v* phenol. Fungal colonies were examined under a stereomicroscope and sexual structures were transferred to Petri dishes (DELTALAB, Barcelona, Spain) containing oatmeal agar (OA; oatmeal flakes, 30 g; agar-agar, 20 g; distilled water, 1 L, homemade) using a sterile needle, and incubated at 15, 25, and 35 °C.

### 2.2. Phenotypic Study

For cultural characterization, isolates were cultured for up to 30 d on OA, potato-carrot agar (PCA; grated potatoes, 20 g; grated carrot, 20 g; agar-agar, 20 g; distilled water, 1 L), and potato dextrose agar (PDA; Pronadisa, Madrid, Spain) at 5, 10, 15, 25, 30, 35, and 40 °C. Color notations in parentheses are from Kornerup and Wanscher [14]. Fertile fungal structures were mounted and measured in water and in lactic acid. Photomicrographs were obtained with a Zeiss (Jena, Germany) Axio Imager M1 light-field microscope with an Olympus DP10 digital camera (Soft Imaging System GMBH, Münster, Germany). The scanning electron microscope techniques used were described previously by Figueras and Guarro [15] with some modifications [16], and micrographs were taken with a Jeol JSM 840 (JEOL Ltd., Tokyo, Japan) at 15 keV.

### 2.3. Molecular Study

DNA of the fungal isolates (Table 1) was extracted and purified directly from colonies according to the Fast DNA Kit protocol (MP Biomedicals, Solon, Ohio, USA). The amplification of the ITS, D1−D3 domains of the LSU, *rpb2,* and *tub2* was performed according to White et al. [17] (ITS), Vilgalys and Hester [18] (LSU), and Miller and Huhndorf [11] (*rpb2* and *tub2*). The sequences of these amplicons were obtained using the protocol of the Big Dye-Deoxy Terminator Cycle Sequencing Kit. PCR products were purified and sequenced at Macrogen Europe (Amsterdam, The Netherlands) with a 3730XL DNA analyzer (Applied Biosystems). Consensus sequences were obtained using SeqMan (version 7.0.0; DNASTAR, Madison, WI, USA). The phylogenetic analysis was carried out based on the combination of the four loci sequences (ITS, LSU, *rpb2,* and *tub2*) of our isolates and selected members belonging to the Sordariales, with *Camarops amorpha* SMH 1450 as an outgroup. Individual gene phylogenies were checked for conflicts before the four gene datasets were concatenated [19,20]. Each locus was aligned separately using MAFFT v. 7 [21] and manually adjusted in MEGA v. 6.06 [22]. The maximum-likelihood (ML) and Bayesian inference (BI) methods were used in a phylogenetic analysis as described by Hernández-Restrepo et al. [23]. The best evolutionary model for each sequence dataset was calculated using MrModeltest v. 2.3 [24]. Bootstrap support (bs) ≥70 and posterior probability values (pp) ≥ 0.95 were considered significant [25]. The sequences generated in this study are deposited in GenBank (Table 1) and the alignments used in the phylogenetic analysis are deposited in TreeBASE (S17160).

## 3. Results

The lengths of the individual alignments used in the combined dataset were 685 bp (ITS), 897 bp (LSU), 984 bp (*rpb2*), and 618 bp (*tub2*), and the final total alignment was 3184 bp. The most likely tree obtained from the RAxML analysis of the combined dataset is shown in Figure 1 and Figure 2. It agreed with the topology of the 95% majority-rule consensus tree generated by the Bayesian analysis. In the combined phylogenetic tree (Figure 1 and Figure 2), *Diplogelasinospora* spp. and the members of Chaetomiaceae, Sordariaceae, and Podosporaceae formed four monophyletic well-supported clades (clade I, 100% bs/1 pp; clade II, 84% bs/- pp; clade III, 100% bs/1 pp; and clade IV, 100% bs/1 pp, respectively). The rest of the Lasiosphaeriaceae s. lat. were distributed in five other monophyletic clades. The type species of the genus *Lasiosphaeria* (*L. ovina*) was placed into a well-supported clade (clade V, 95% bs/1 pp) together with other spp. of the genus, species of *Cercophora, Podospora*, *Zopfiella, Anopodium ampullaceum, Bellojisia rhynchostoma*, and *Corylomyces selenospora*. A second clade (clade VI, unsupported) included the species of the genera *Apodospora*, *Bombardia*/*Bombardioidea*, *Fimetariela*, and *Cercophora scortea*, and several species of *Podospora* and *Zopfiella*. The third clade (clade VII, 86% bs/1 pp) only included the species of *Lasiosphaeris* and *Zygospermella insignis*. The fourth clade (clade VIII, 80% bs/1 pp) included the species of *Schizothecium* and those of *Echria*, *Immersiella*, *Rinaldiella*, *Jugulospora*, *Strattonia*, and *Zygopleurage*, plus several species of *Arnium*, *Apiosordaria*, *Cercophora*, *Triangularia*, and *Zopfiella*. Finally, the fifth clade (clade IX, 100% bs/1 pp) included different species of *Arnium*, *Cercophora*, *Podospora*, *Triangularia*, and *Zopfiella*, none of them corresponding to the type species of the mentioned genera, and the strain CBS 137295, which was placed in a terminal branch. CBS 137295 displayed a nucleotide identity lower than 95% with respect to the other taxa included in our phylogenetic study.

### Taxonomy

As the species of *Diplogelasinospora* were located in a well-supported clade (clade I, 100% bs/1 pp; Figure 1) distinct from the Sordariaceae (clade III), and because the morphology and ornamentation (i.e., inwardly pitted upper and lower cell) of the ascospores are unique in the Lasiosphaeriaceae s. lat., we propose the new family Diplogelasinosporaceae to accommodate this genus.

**Diplogelasinosporaceae Y. Marin & Stchigel, fam. nov.** MycoBank MB835495.

*Etymology*: Named for *Diplogelasinospora*, the type genus of this family.

*Type genus*: *Diplogelasinospora* Cain, Can. J. Bot. 39: 1669. 1961.

Ascomata non-ostiolate, immersed or superficial, scattered or in groups, dark brown to nearly black, spherical, non-stromatic, sometimes covered by hyphae-like setae; ascomatal wall membranaceous to slightly coriaceous, *textura angularis*, occasionally cephalothecoid or outer layer of *textura intricata*, dark brown, opaque. Asci unitunicate, eight-spored, cylindrical to cylindrical-clavate, short stipitate, lacking a distinct apical ring, arranged in parallel fascicles, evanescent. Paraphyses present, cylindrical to moniliform, septate and sometimes constricted at the septa. Ascospores at first hyaline, aseptate, ellipsoidal, becoming transversely septate and two-celled, one cell remaining hyaline, one becoming black and opaque, wall smooth or pitted showing an endodentate endosporium, sometimes with a thin gelatinous sheath. Asexual morph absent or present. Conidiophores lacking or not distinctly differentiated from vegetative hyphae. Conidia arthric, terminal or intercalary, hyaline, oblong to cylindrical, with both ends truncated, smooth-walled, sometimes sliding into a slimy mass; conidia blastic, borne singly and laterally on aerial hyphae, hyaline, ovate to elongate, occasionally subspherical to somewhat elongate, smooth-walled.

Notes: This family includes only species of the genus *Diplogelasinospora*, which was originally in Neurosporaceae (syn. Sordariaceae) based on the similar ascospore ornamentation of the species belonging to the genus *Neurospora* (formerly *Gelasinospora*) [37]. Cai et al. [33] observed that this genus did not group in the Sordariaceae and demonstrated that it was more related to the Lasiosphaeriaceae. In our phylogenetic study, this genus formed a well-supported clade distinct from the other families of the Sordariales. Diplogelasinosporaceae can be distinguished from other members of the Lasiosphaeriaceae s. lat., Naviculisporaceae, Schizotheciaceae, and Podosporaceae by the presence of an asexual morph characterized by the production of arthroconidia. A similar asexual morph has been observed in Chaetomiaceae and Sordariaceae. However, the latter ones produce one-celled ascospores, while ascospores are two-celled in Diplogelasinosporaceae.

The type species of the genus *Lasiosphaeria* (*L. ovina*) was placed into a well-supported clade (clade V, 95 bs/1 pp) together with other spp. of that genus, *Corylomyces selenospora* and *Bellojisia rhychostoma*, and several species of *Anopodium*, *Cercophora*, *Podospora*, and *Zopfiella*. As the rest of the members of Lasiophaeriaceae fall into other distinct clades, we restrict the circumscription of the family Lasiosphaeriaceae as described below.

**Lasiosphaeriaceae Nannf., Nova Acta R. Soc. Scient. upsal., Ser. 4 8(no. 2): 50, 1932, emend. Y. Marin & Stchigel.** MycoBank MB80930.

*Type genus: Lasiosphaeria* Ces. & De Not., Comment. Soc. Crittog. Ital. 1 (fasc. 4): 229. 1863.

Ascomata mostly ostiolate, immersed to superficial, scattered or aggregated, olivaceous yellow to dark brown, pyriform, globose, subglobose or ovoid, covered with white to variously colored (but not dark) tomentum below the neck, or with hyaline, light brown or olive, sometimes rigid, septate hairs on the neck; neck short or long, black or nearly so, cylindrical to conical; non-ostiolate ascomata globose, covered with yellowish to olivaceous-brown hyphae-like hairs; ascomatal wall membranaceous to coriaceous, *textura angularis* to *textura intricata*, weakly cephalothecoid in non-ostiolate ascomata. Paraphyses absent or present, filiform. Asci unitunicate, four- or eight-spored, cylindrical to cylindrical-clavate or clavate, short- or long-stipitate, with an apical ring, distinct or inconspicuous, or with an apical differentiation refractive and non-amyloid, usually with a subapical globule, or without apical differentiations. Ascospores uniseriate or biseriate, one- or two-celled. One-celled ascospores hyaline to yellowish, sometimes turning pale brown with age, or at first hyaline and then becoming dark brown, cylindrical, irregularly ellipsoidal, reniform, or navicular, mostly sigmoid, geniculate or curved, sometimes with an umbonate germ pore, with or without gelatinous appendages, one end may become swollen and turn brown. Two-celled ascospores at first hyaline and one-celled, becoming transversely septate; upper cell olivaceous to dark brown, ellipsoidal or vermiform to clavate, with an apical or subapical germ pore, sometimes later developing a transversal septum; lower cell hyaline, cylindrical, collapsing or not; gelatinous cauda sometimes present, lash-like, attached at both ends. Asexual morph absent or present. Conidiophores reduced to conidiogenous cells. Conidiogenous cells phialidic, monophialidic, or sometimes polyphialidic, produced laterally or terminally, delimited by a basal septum, obclavate to lageniform, hyaline to pale brown, constricted below the collarette. Conidia enteroblastic, aggregated in slimy heads, hyaline, pyriform, truncate at base; conidia holoblastic absent or present, produced directly from hyphae or lateral branches not delimited by a basal septum and without collarette, hyaline, subglobose, pyriform, or obclavate.

Notes: Lasiosphaeriaceae s. str. was divided in two main clades, one (100% bs/1 pp) encompassing only species of the genus *Lasiosphaeria* (including the type species of the genus, *L. ovina*, and *L. glabrata*, *L. lanuginosa*, *L. rugulosa* and *L. sorbina*), and a second one (100% bs/1 pp) containing the type species of *Zopfiella* and other taxa producing ascospores with a septate upper cell (with the exception of *Anopodium ampullaceum* [38]). Further studies are needed to clarify if the genus *Zopfiella* should be restricted to this second clade. This family is characterized by the production of (mostly) ostiolate ascomata with a tomentose ascomatal wall or bearing septate hairs on or below the neck. The family includes taxa producing one- and two-celled ascospores. Within the first group, a high morphological variability of the ascospores can be found, those of *Lasiosphaeria* being cylindrical and sigmoid, geniculate or curved [10] vs. reniform to navicular in *Bellojisia rhynchostoma* and *Corylomyces selenospora* [28,31].

In order to accommodate three new genera, i.e., *Areotheca*, *Naviculispora*, and *Rhypophila*, as well as some species of *Arnium, Triangularia*, and *Zopfiella*, which form a monophyletic clade (clade IX, 100% bs/1 pp) distinct from Lasiosphaeriaceae s. str., we propose the erection of the new family Naviculisporaceae as follows.

**Naviculisporaceae Y. Marin & Stchigel, fam. nov.** MycoBank MB835496.

*Etymology*: Named after *Naviculispora*, the type genus of this family.

*Type genus*: *Naviculispora* Stchigel, Y. Marín, Cano & Guarro.

Ascomata ostiolate, less frequently non-ostiolate, immersed or superficial, scattered or aggregated, brown to dark brown or black, globose to subglobose, pyriform or ovate, glabrous or covered with flexous or stiff hairs; neck short or long, dark brown to blackish, conical or cylindrical, papillate, rarely glabrous, mostly with elongate tuberculated projections at the base or covered with small black papillae; ascomatal wall membranaceous, pseudoparenchymatous, semi-translucent, yellowish to light brown or olivaceous brown, *textura angularis* or irregularly-shaped cells, or blackish brown, opaque, areolate, cracking in polyhedral plates when crushed. Paraphyses absent or present. Asci unitunicate, 8- to 256-spored, cylindrical or clavate, evanescent, long- or short-stipitate, without apical ring, or apical ring indistinct or distinct, small, subapical globulus or chamber absent or present. Ascospores biseriate or irregularly arranged into the asci, mostly two-celled. One-celled ascospores at first hyaline, becoming ochraceous to brown or blackish brown when mature, ellipsoidal, with somewhat pointed ends, smooth-walled, provided at each end with a germ pore and a lash-like gelatinous cauda. Two-celled ascospores at first one-celled and hyaline, clavate, cylindrical or spatuliform, sometimes vermiform, becoming transversely septate and two-celled; upper cell olivaceous or brown to dark brown, navicular, conical, ellipsoidal, ellipsoidal-fusiform, limoniform, pyriform or obclavate, smooth-walled, septate, with an apical or subapical germ pore, sometimes with one or more transverse septa; lower cell hyaline or pale brown, rarely dark brown, hemispherical, cylindrical, cylindrical-obclavate, or cylindrical-conical, thick- or thin-walled, sometimes with the same length or longer than the upper cell, collapsing or not with age, rarely with 3-5 septa; several secondary appendages sometimes present, small, at the base; apical gelatinous cauda absent or present, lash-like, fibrillate or lamellate. Asexual morph absent or present. Conidiphores reduced to conidiogenous cells or sympodially proliferating. Conidiogenous cells phialidic. Conidia holoblastic, sometimes sessile, hyaline to subhyaline, ellipsoidal to subspherical, obovoid or clavate.

Notes: The family shows a high morphological variability: *Arnium caballinum*, *A. japonense*, and *A. mendax* produce one-celled ascospores [9,39], while in the rest of the species of the family, these are two-celled. Taxa with both kinds of ascospores are also observed in the family Lasiosphaeriaceae s.str. as well as in Podosporaceae and Schizotheciaceae. The ascomata are mostly ostiolate, except in *Z. marina*, *Z. pilifera*, and *Z. submersa*, in which they are non-ostiolate [40,41,42]. In our phylogenetic study, these three species with non-ostiolate ascomata grouped in a well-supported clade (100 bs/1 pp) together with *T. mangenotii*, which produces ostiolate ascomata [43], suggesting that these could represent a new genus. Further molecular and phenotypic studies are needed to confirm this hypothesis. The ascomatal wall is mostly membranaceous, except in the new genus *Areotheca* (introduced below), which is characterized by an areolate ascomatal wall. Among the Sordariales, this type of ascomatal wall is only seen in two species of *Cladorrhinum*, i.e., *Cl. coprophilum* and *Cl. tomentosum*. It is also found in a few other *Cercophora* species, i.e., *C. caerulea*, *C. septentrionalis*, and *C. sylvatica,* but their walls are not as carbonaceous as in *Areotheca*, in which the fully developed carbonaceous wall breaks into polyhedral plates when crushed.

Within the Naviculisporaceae, reference strains of *C. ambigua* and *C. areolata* formed a well-supported clade (100 bs/1 pp; Figure 2). Both species are characterized by a well-developed carbonaceous, areolate ascomatal wall that cracks into polyhedral plates when crushed (cephalothecoid). The new genus *Areotheca* is proposed here to accommodate these two species.

***Areotheca* Y. Marín & Stchigel, gen. nov.** MycoBank MB835497. (Figure 3A–D).

*Type species*: *Areotheca ambigua* (R. Hilber) Y. Marín & Stchigel.

*Etymology*: The name refers to the areolate ascomatal wall.

Ascomata ostiolate, scattered or aggregated, superficial, ovoid, glabrous or covered with flexuous, septate hairs; neck conical, ridged, carbonaceous; ascomatal wall pseudoparenchymatous, blackish brown, carbonaceous, opaque, areolate, cephalothecoid, 2–3 layered; outer ascomatal wall cells prismatic; intern layers composed of flattened cells. Paraphyses filiform-venticrose. Asci unitunicate, eight-spored, narrowly clavate or cylindrical, non-amyloid, apical ring thickened, with or without subapical globulus. Ascospores biseriate or multiseriate, at first one-celled, hyaline, cylindrical, vermiform, slightly sigmoid or bent in one end, later swelling above, becoming transversely uniseptate; upper cell brown, ellipsoidal to pyriform, more or less equilateral, smooth-walled, truncate at base, with a subapical germ pore, sometimes with one or more transverse septum; lower cell hyaline to pale brown, cylindrical, geniculate below, collapsing with age, without septa or 3–5-septate; gelatinous cauda absent or present, lash-like, attached at one or both ends of the ascospore. Asexual morph absent or present. Conidiophores reduced to conidiogenous cells. Conidiogenous cells phialidic, hyaline, arising directly from ascospores, normally from the lower cell, rarely branched. Conidia hyaline, globose, smooth-walled.

Notes: The main distinctive feature of *Areotheca* is its carbonaceous, areolate, cephalothecoid ascomatal wall. In this regard, other taxa in the Sordariales produce this type of ascomatal wall: *Cladorrhinum coprophilum* (syn. *C. coprophila*) shows indistinct areoles, whereas they are more obvious in *C. caerulea*, *C. septentrionalis*, *C. silvatica*, and *T. striata* (syn. *C. striata*). However, these particular structures are fully developed and carbonaceous in *Ar. areolata*. Another important morphological feature is the septation of the upper cell, present only in the genus *Naviculispora* and some species (*Bellojisia rhynchostoma*, *C. sulphurella, Corylomyces selenospora, Podospora dydima*, and *Zopfiella tabulata*) included in the Lasiosphaeriaceae s. str., and *Zopfiella pleuropora* (of clade VI). Finally, *Areotheca* is distinguished from other genera in the family by its ascospores with a basal cell up to twice as long as the upper cell. The two species of *Areotheca* are mainly differentiated by the size of their ascospores (the upper cell measures 20–27 × 8–11 µm in *Ar. ambigua* and 15–18 × 7–10 µm in *Ar. areolata*; the lower cell measures 35–55 × 6–7 µm in *Ar. ambigua* and 25–32 × 4–5 µm in *Ar. aerolata*), and by the presence of several septa in the lower ascospore cell of *Ar. ambigua*, absent in *Ar. areolata*.

***Areotheca areolata* (N. Lundq.) Y. Marín, A.N. Mill. & Stchigel, comb. nov.** MycoBank MB 835498. (Figure 3A–D).

*Basionym*: *Cercophora areolata* N. Lundq., Symb. Bot. Upsal. 20: 104. 1972.

***Areotheca ambigua* (Sacc.) Y. Marín & Stchigel, comb. nov.** MycoBank MB835681.

*Basionym: Lasiosphaeria hispida subsp.* ambigua *Sacc., Michelia 1: 46. 1877*.

*Synonym: Lasiosphaeria ambigua* (Sacc.) Sacc., Sylloge Fungorum 2: 197. 1883.

*Bombardia ambigua* (Sacc.) G. Winter, Rabenhorst’s Kryptogamen-Flora, Pilze - Ascomyceten, Edn 2 1: 236. 1885.

*Lasiosordaria ambigua* (Sacc.) Chenant., Bull. Soc. Mycol. France 35: 79. 1919.

*Cercophora ambigua* (Sacc.) R. Hilber, in Hilber & Hilber, Z. Mykol. 45: 212. 1979.

As the strain CBS 13729 was placed in a clade far from Lasiosphaeriaceae s. str., and because it differs from all other members of Lasiosphaeriaceae s. lat. by the production of navicular ascospores whose upper cell is dark brown and septate and the lower cell is pale brown and non-collapsing with age, it is proposed as belonging to the new genus *Naviculispora* as follows:

***Naviculispora* Stchigel, Y. Marín, Cano & Guarro, gen. nov.** MycoBank MB 812135. (Figure 4A–H).

*Etymology*: From Latin *navicularibus*-, navicular, and -*sporarum*, spore, referring to the shape of the ascospores.

*Type species*: *Naviculispora terrestris* Stchigel, Cano, Y. Marín & Guarro.

Ascomata ostiolate, brown, pyriform, covered with septate hyphae-like hairs; neck short, dark brown, papillate; ascomatal wall membranaceous, *textura angularis*. Asci unitunicate, eight-spored, cylindrical, short stipitate, with a small apical ring, evanescent. Ascospores at first one-celled, hyaline, clavate, becoming transversely septate and two-celled; upper cell dark brown, navicular, septate, with a subapical to lateral germ pore; lower cell pale brown, cylindrical-conical, thick-walled, not collapsing with age; gelatinous cauda absent. Conidia holoblastic, hyaline to subhyaline, ellipsoidal to obovoid or clavate, sessile or less commonly on sympodially proliferating conidiophores.

Notes: *Naviculispora* spp. can be easily distinguished from the two phylogenetically related species of *Arnium* (*Arnium caballinum* and *A. japonense*) by the production of two-celled ascospores (one-celled in *Arnium* spp.). *Areotheca*, as well as *Naviculispora*, produces two-celled ascospores with a septum in the upper cell. However, these taxa differ in the nature of the ascomatal wall, being membranaceous in *Naviculispora* and cephalothecoid in *Areotheca*. As in *Naviculispora* spp., the upper cell of the ascospores in *Bellojisia rhynchostoma*, *Cercophora sulphurella*, *Corylomyces selenospora*, *Podospora dydima*, and *Zopfiella tabulata* is septate; however, the lower cell of the ascospores is pale brown to brown, thick-walled, and non-collapsing with age in *Naviculispora* spp., whereas, in the other taxa, it is hyaline (or nearly so) and collapses very soon, except for *C. sulphurella*. *Zopfiella attenuata* and *Z. pleuropora*, both located in a non-supported main clade (clade VI, Figure 1), also produce ascospores with a septate upper cell, but differ from the Naviculisporaceae by the production of non-ostiolate ascomata.

***Naviculispora terrestris* Stchigel, Y. Marín, Cano & Guarro, sp. nov.** MycoBank MB 812136. (Figure 4A–H).

*Etymology*: Referring to the substrate from which the fungus has been isolated.

*Type material*: ARGENTINA: Tucumán province, Tafí del Valle, -26.8667, -65.6833, from soil, 17-V-2000, A. M. Stchigel, J. Cano, J. Guarro. (holotype CBS H-2159; ex-type cultures CBS 137295 = FMR 10060).

Mycelium composed of subhyaline to brown, septate, smooth-walled, branched hyphae, 1.5–4 µm wide. Ascomata ostiolate, superficial or immersed, scattered to aggregated, pale brown to brown, pyriform, 210–410 × 160–380 µm, covered with brown, septate hyphae-like hairs, occasionally with 2 necks; necks short, brown to dark brown, cylindrical to conical, papillate, 35–70 µm long, 62.5–105 µm wide; ascomatal wall membranaceous, 10–20 µm thick, composed of up to 10 layers of flattened cells of 2–10 µm in diam, *textura angularis*. Asci unitunicate, eight-spored, cylindrical, evanescent, 100–150 × 15–20 µm, short stipitate, with small apical ring. Ascospores uniseriate to biseriate, at first one-celled, hyaline, clavate, becoming transversely septate and two-celled; upper cell dark brown, navicular, 20–29 × 9–13.5 µm, truncate at the base, septate, with a subapical to apical germ pore of 0.5–1 µm; lower cell pale brown, cylindrical-conical, thick-walled, 10–14(–17) × (2–)3–4 µm, not collapsing; gelatinous cauda absent. Asexual morph present. Conidia holoblastic, 3–8 × 2–3 µm, hyaline to subhyaline, ellipsoidal to ovoid or clavate, with a small truncate base, sessile, produced laterally and terminally on the hyphae, or less commonly on sympodially proliferating conidiophores.

Culture characteristics: Colonies on PCA attaining a diam. of 40–47 mm in 14 d at 25 °C, velvety, slightly lobulate, margins fringed, radially zonate, greyish-brown (M 5D3 to 5E3) and brown to dark brown (6E3 to 6F3) at center; reverse greyish-yellow to olive brown (4B4 to 4D4). Colonies on OA attaining a diam. of 35–38 mm in 14 d at 25 °C, velvety, margins regular to slightly arachnoid, grey to greyish-brown (M 8F1 to 8F3); reverse grey to greyish brown (M 8F1 to 8F3). Ascomata produced after at least two months. Maximum and minimum temperatures of growth, 5 and 30 °C, respectively. Optimal temperature 25 °C.

Reference strains of *P. cochleariformis*, *P. decipiens, P. myriaspora*, and *P. pleiospora*, classified by Lundqvist [9] into the section *Rhypophila* of the genus *Podospora*, formed a well-supported clade (100% bs/ 1 pp) in the Naviculisporaceae. All four species are characterized by the production of an ascomatal neck with elongate tuberculated projections at the base, asci usually with more than eight ascospores, and ascospores with a lower cell usually longer than the upper cell. Based on their morphologically distinctive features and our phylogenetic results, we erect the genus *Rhypophila* as follows:

***Rhypophila* Y. Marín, A.N. Mill. & Guarro, gen. nov.** MycoBank MB 812130. (Figure 5A–K).

*Etymology*: Referring to the name of the section *Rhypophila* of the genus *Podospora*, proposed by Lundqvist (1972).

*Type species*: *Rhypophila myriospora* (P. Crouan & H. Crouan) Y. Marín, A.N. Mill. & Guarro.

Ascomata ostiolate, semi-immersed or superficial, scattered or aggregated, pyriform, glabrous or covered with flexous or stiff hairs; neck long, conical or cylindrical, with blackish, obtuse, straight or curved, elongate tuberculated projections at the base; ascomatal wall membranaceous, semi-translucent, yellowish to light brown. Paraphyses absent or present. Asci unitunicate, 8- to 128-spored, clavate, long or short stipitate, apical ring absent or indistinct. Ascospores biseriate or irregularly arranged, at first one-celled, hyaline, cylindrical or spatuliform, clavate, later swelling above, becoming transversely uniseptate and two-celled; upper cell dark brown, ellipsoidal to ellipsoidal-fusiform, with an apical or subapical germ pore; lower cell hyaline, cylindrical to cylindrical-obclavate, same length or longer than upper cell, frequently collapsing; secondary appendages small, at the base of the lower cell; apical gelatinous cauda absent or present, fibrillate or lamellate. Asexual morph absent.

Notes: *Rhypophila* includes species formerly classified in a section with the same name in the genus *Podospora* [9]. Chang et al. [12] also noted that *Podospora* species of the section *Rhypophila* grouped together in the phylogenies based on ITS and a fragment of the glyceraldehyde-3-phosphate dehydrogenase (*gpd*) gene. Although the phylogenetic distance between *R. cochlearifomis/R. myriaspora* and *R. decipiens/R. pleiospora* suggests that these could represent two different genera, we decided to include all these species in *Rhypophila* based on the morphological resemblances. Further studies including more taxa are needed to clarify it. The genus *Rhypophila* is characterized by the production of ostiolate ascomata with a neck bearing elongated tuberculated projections at the base, asci normally containing more than eight ascospores, and ascospores with a lower cell as long as, or longer than, the upper cell. Ornamentation in the upper part of the ascomata is also present in *Pseudoechria, Pseudoschizothecium*, and *Schizothecium*, but these three genera, classified in the new family Schizotheciaceae (clade VIII, Figure 2), differ in the nature of the ornamentation (see Notes on *Pseudoechria*).

***Rhypophila cochleariformis* (Cailleux) Y. Marín, A.N. Mill., Guarro, comb. nov.** MycoBank MB 812131.

*Basionym*: *Podospora cochleariformis* Cailleux, Cah. Maboké 7: 100. 1969.

***Rhypophila decipiens* (G. Winter ex Fuckel) Y. Marín, A.N. Mill., Guarro, comb. nov.** MycoBank MB 812132. (Figure 5E,G–I)

*Basionym*: *Sordaria decipiens* G. Winter, Abh. Naturf. Ges. Halle 13: 28. 1873.

*Synonym*: *Podospora decipiens* (G. Winter ex Fuckel) Niessl, Hedwigia 22: 156. 1883.

***Rhypophila myriospora* (P. Crouan & H. Crouan) Y. Marín, A.N. Mill., Guarro, comb. nov.** MycoBank MB 812133. (Figure 5B,D,K).

*Basionym*: *Sordaria myriospora* P. Crouan & H. Crouan, Florule Finistère (Paris): 22. 1867.

*Synonym*: *Podospora myriospora* (P. Crouan & H. Crouan) Niessl, Hedwigia 22: 156. 1883.

***Rhypophila pleiospora* (G. Winter) Y. Marín, A.N. Mill. & Guarro, comb. nov.** MycoBank MB 812134. (Figure 5A,C,F,J).

*Basionym*: *Sordaria pleiospora* G. Winter, Abh. Naturf. Ges. Halle 13: 13. 1873.

*Synonym*: *Podospora pleiospora* (G. Winter) Niessl, Hedwigia 22: 156. 1883.

Key to species of *Rhypophila*.
1. Asci 8-spored*R. decipiens*1. Asci 16 to 128-spored22. Asci 128-spored; upper and lower cells each shorter than 20 µm; absence of cauda and gelatinous appendages*R. cochleariformis*2. Asci 16-, 32- or 64-spored; upper and lower cells each longer than 20 µm; fibrillate upper cauda and gelatinous appendages in the lower cell33. Asci 16 or 32 spored; upper cell 23–34 × 14–19 µm*R. myriospora*3. Asci 64 spored; upper cell 25–37 × 18–23 µm*R. pleiospora*

The Podosporaceae (100% bs/1 pp; clade IV, Figure 1) formed three fully supported clades (100% bs/1 pp) representing the genera *Cladorrhinum*, *Podospora*, and *Triangularia*. The reference strains of *A. tomentosum* and *C. coprophila* were nested in the *Cladorrhinum* clade. Consequently, the description of *Cladorrhinum* is emended, and new combinations for *A. tomentosum* and *C. coprophila* are proposed as follows:

***Cladorrhinum* Sacc. & Marchal, Bull. Soc. Roy. Bot. Belgique 24: 64. 1885. emend. Y. Marin & Stchigel.** MycoBank MB7678.

*Type species: Cladorrhinum foecundissimum* Sacc. & Marchal.

Ascomata ostiolate or non-ostiolate, superficial to immersed, solitary or aggregated, globose to subglobose, ovoid, conical, obpyriform or broadly obpyriform, covered with hypha-like ascomatal hairs or by aerial mycelium, or a dense tomentum of thick-walled hairs; ascomatal wall membranaceous, semi-translucent to opaque, *textura angularis*, *textura intricata* or *textura epidermoidea* in surface view, or sometimes composed of patches or indistinct areoles; neck cylindrical, black, opaque, carbonaceous, often covered by rigid, septate hairs. Paraphyses absent or present, filiform-vetricose. Asci unitunicate, eight-spored, cylindrical, narrowly clavate, pyriform, obovoid or fusiform, short- or long-stipitate, without or with a thin or thick apical ring, evanescent. Ascospores uniseriate into the asci, one- or two-celled. One-celled ascospores ochraceous or olivaceous brown to dark brown, ellipsoidal to broadly fusiform, ovoid or obovoid, smooth-walled, with a germ pore at each end or with an apical germ pore, a diffuse sheath surrounding the whole spore sometimes present; gelatinous cauda absent or present, attached to each end. Two-celled ascospores biseriate or triseriate, at first hyaline, one-celled, vermiform, slightly sigmoid or geniculate only below, smooth-walled, then swelling above, becoming transversely septate and two-celled; upper cell brown, ellipsoidal, truncate at the base, conical above with an apical germ pore; lower cell cylindrical, collapsing; gelatinous cauda attached to each end. Asexual morph absent or present. Conidiophores micronematous, reduced to conidiogenous cells. Conidiogenous cells mostly intercalary, occasionally terminal, originating as a lateral or terminal peg-like structure with a flaring collarette. Conidia enteroblastic, one-celled, hyaline, broad obovoid, ellipsoidal or subglobose, usually with a truncated base and a rounded apex, smooth-walled.

Notes: *Cladorrhinum* previously included species that only produced an asexual morph characterized by the production of one-celled conidia in slimy masses from intercalary enteroblastic conidiogenous cells with lateral openings (integrated phialides; adelophialides). Fertile hyphae can develop more or less branched conidiophores, and terminal and lateral phialides and microsclerotia can sometimes also be observed in culture [44,45]. Recently, Wang et al. [6] maintained in the genus only the type species, *Cl. foecundissimum*, based on sequence data. They also observed that two species previously identified as *Thielavia hyalocarpa* and *Th. intermedia*, clustered together with *Cl. foecundissimum*. Therefore, the description of *Cladorrhinum* was emended to incorporate the presence of a sexual morph characterized by non-ostiolate ascomata and one-celled ascospores. In our phylogenetic study, *C. coprophila*, which produces ostiolate ascomata and two-celled ascospores, as well as *A. tomentosum*, which forms ostiolate ascomata and one-celled ascospores, also grouped in the *Cladorrhinum* clade. Therefore, the new combinations are proposed here and the description of the genus is newly emended to incorporate these new morphological characteristics. In addition, a reference strain of *A. olerum*, CBS 120012, occurred in a well-supported clade (100% bs/0.98 pp) together with the type strain of *Cl. foecundissimum*, suggesting that the former species is the sexual stage of the asexual *Cl. foecundissimum*; therefore, their synonymy is proposed here. *Cladorrhinum*, Podospora, and Triangularia present the same asexual morph. However, these genera differ among the sexual morphs. The ascospores are mostly one-celled in *Cladorrhinum*, whereas they are mostly two-celled in *Podospora* and *Triangularia*. Ostiolate and non-ostiolate ascomata can be found in all three genera, suggesting that the ascomatal morphology in Podosporaceae is not phylogenetically informative. Cai et al. [46] had already observed this high variability and reached the same conclusion. Sequence data are necessary to properly delineate these genera, which display a high variability in the ascomata and ascospore morphology, with many overlapping features among them. *Cladorrhinum coprophilum* and *Cl. tomentosum* produce ascomata with a wall composed of patches or indistinct areoles, a feature also observed in *T. striata*. However, this latter species can be distinguished from the former two by its tuberculate projections on the neck giving a striate appearance.

***Cladorrhinum coprophilum* (Fr.) Y. Marin, A.N. Mill. & Stchigel, comb. nov.** MycoBank MB 835499. (Figure 6A–G).

*Basionym: Sphaeria coprophila* Fr., in Kunze & Schmidt, Mykol. Hefte (Leipzig) 2: 38. 1823.

*Synonym: Cercophora coprophila* (Fr.) N. Lundq., Symb. Bot. Upsal. 20: 95. 1972.

Other synonyms in [9].

Because the reference strain of *Arnium olerum* formed a well-supported terminal lineage with *Cladorrhinum foecundissimum* (100% bs/0.98 pp), the synonymy of *A. olerum* under *C. foecundissimum* is proposed here.

**Cladorrhinum foecundissimum Sacc. & Marchal, Bull. Soc. R. Bot. Belg. 24: 64. 1885**. MycoBank MB119723. (Figure 7A–E).

*Synonyms: Sphaeria olerum* Fr., Elench. Fung. (Greifswald) 2: 98. 1828.

*Arnium olerum* (Fr.) N. Lundq. & J.C. Krug, Symb. Bot. Upsal. 20: 212. 1972.

For other synonyms see reference [9].

***Cladorrhinum tomentosum* (Speg.) Y. Marin & Stchigel, comb. nov.** MycoBank MB835500.

*Basionym: Hypocopra tomentosa* Speg., Anales Soc. Ci. Argent.10: 136. 1880.

*Synonym: Arnium tomentosum* (Speg.) N. Lundq. & J.C. Krug, Symb. Bot. Upsal. 20: 216. 1972.

*Sordaria tomentosa* (Speg.) Sacc., Syll. Fung. (Abellini) 1: 236. 1882.

*Podospora tomentosa* (Speg.) Niessl, Hedwigia 22: 156. 1883.

Key to species of *Cladorrhinum*.
1. Two-celled ascospores*Cl. coprophilum*1. One-celled ascospores22. Non-ostiolate ascomata32. Ostiolate ascomata43. Ascospores (22.5–)24.5–31(–36) × (11.5–)13–15.5(–16) μm*Cl. hyalocarpum*3. Ascospores 12–17 × 9–12 μm*Cl. foecundissimum*4. Ascomatal wall not areolate, ascospores 40–52 × 25–32 µm*Cl. olerum*4. Ascomatal wall areolate, ascospores 47–58 × 22–28(–30) µm*Cl. tomentosum*

The type strains of *Apiosordaria sacchari* and *Ap. striatispora*, and a reference strain of *C. costaricensis*, formed independent lineages within *Podospora*. Therefore, the new combinations are proposed here, and the description of *Podospora* is emended:

***Podospora* Ces., Hedwigia 1: 103. 1856. emend. Y. Marin & Stchigel**. MycoBank MB4284.

*Type species: Podospora fimicola* (Corda) Ces.

Ascomata usually superficial, solitary, ostiolate and pyriform, or less frequently non-ostiolate and globose, covered by erect or flexuous, hyphae-like hairs; neck short, conical; ascomatal wall membranaceous to coriaceous, sometimes pseudo-bombardioid and containing a central layer composed of thick-walled, gelatinized cells. Paraphyses absent or present. Asci unitunicate, eight-spored, clavate or cylindrical, with an apical ring, usually persistent until ascospore maturity. Ascospores uniseriate or biseriate, two-celled; upper cell olivaceous brown to brown or dark brown, ellipsoidal to ovoid or obovoid, or five-angled in side view, smooth-walled, spinulose (spines up to 3 μm long), or with several longitudinal striations under light microscopy or by SEM exhibiting several longitudinal ribs that result in a coalescence of warts, with an apical germ pore; lower cell hyaline to pale brown, clavate or cylindrical, smooth-walled; apical and basal mucilaginous appendages mostly present. Conidiophores micronematous, reduced to conidiogenous cells. Conidiogenous cells usually intercalary, originating from a lateral peg-like structure with a flaring collarette. Conidia blastic, single-celled, hyaline, smooth-walled, globose or ellipsoidal to elongate, usually with a truncated base and a rounded apex.

Notes: *Podospora* was one of the largest and morphologically most diverse genera in Lasiosphaeriaceae [9]. Recently, this was reduced to the type species, *P. fimicola*, and the asexual species, *P. bulbillosa*, previously known as *Cl. bulbillosum* [6]. In the same study, it was designated as the type genus of the family Podosporaceae, which includes only two genera, *Cladorrhinum* and *Triangularia*. For morphological comparison, see Notes of *Cladorrhinum*.

***Podospora costaricensis* (G.C. Carroll & Munk) Y. Marín, A.N. Mill. & Stchigel, comb. nov**. MycoBank MB835501. (Figure 8A–G).

*Basionym: Bombardia costaricensis* G.C. Carroll & Munk, Mycologia 56: 80. 1964.

*Synonym: Cercophora costaricensis* (G.C. Carroll & Munk) O. Hilber & R. Hilber, Z. Mykol. 45: 217. 1979.

***Podospora sacchari* (B.M. Robison) Y. Marín & Stchigel, comb. nov**. MycoBank MB835502.

*Basionym: Echinopodospora sacchari* B.M. Robison, Trans. Brit. Mycol. Soc. 54: 320. 1970.

*Synonym: Apiosordaria sacchari* (B.M. Robison) J.C. Krug, Udagawa & Jeng, Mycotaxon 17: 546. 1983.

***Podospora striatispora* (Furuya & Udagawa) Y. Marín & Stchigel, comb. nov**. MycoBank MB835503.

*Basionym: Triangularia striatispora* Furuya & Udagawa, J. Jap. Bot. 51: 406. 1976.

*Synonym: Apiosordaria striatispora* (Furuya & Udagawa) Guarro & Cano, Trans. Brit. Mycol. Soc. 91: 589. 1988.

Key to species of *Podospora*.
1. Only asexual morph present*P. bulbillosa*1. Sexual morph present22. Ascospores smooth-walled32. Ascospores ornamented43. Upper cell 44.5–55 × 27.5–46 μm, lower cell 25–37 × 4.5–6.5(–8) μm*P. fimicola*3. Upper cell 18–22 × 8–9(–10) μm, lower cell 22–27 × 4–5 μm*P. costaricensis*4. Upper cell obovoid, spinulose, 33.2–40.5 × 24.8–31.3 μm*P. sacchari*4. Upper cell five-angled in side view, ornamented with longitudinal ribs, 10–12(–15) × 8–9 μm*P. striatispora*

Because the reference strains of *A. arizonensis*, *C. striata*, and *Z. tetraspora* grouped in the *Triangularia* clade, new combinations are proposed, and the description of *Triangularia* is emended:

***Triangularia* Boedijn, Ann. Mycol. 32: 302. 1934, emend. Y. Marin & Stchigel**. MycoBank MB5534.

*Synonym: Apiosordaria* Arx & W. Gams, Nova Hedwigia 13: 201. 1967.

*Type species: Triangularia bambusae* (J.F.H. Beyma) Boedijn, Ann. Mycol. 32: 302. 1934.

Ascomata ostiolate or non-ostiolate, often superficial, sometimes semi-immersed to immersed, solitary or aggregated, globose to subglobose, ovoid, or obpyriform to ampulliform, glabrous or with hypha-like or seta-like hairs; neck papillate, sometimes with rigid, agglutinated, cylindrical, septate, thick hairs, sometimes striate with 5–8 striations; ascomatal wall membranaceous to coriaceous, translucent, semitranslucent, or opaque, *textura angularis* to *textura globulosa*, sometimes composed of areoles of *textura angularis*. Periphyses absent or present. Paraphyses usually present, hyaline, filiform or filiform-ventricose. Asci unitunicate, four-, eight-, or multi-spored, cylindrical to elongated clavate or fusiform, short or long stipitate, without or with an inconspicuous or thick apical ring, evanescent or persistent until ascospores mature. Ascospores one- or two-celled. One-celled ascospores uniseriate, at first hyaline, becoming yellowish, olivaceous to brown-black, ellipsoidal, smooth-walled, with an apical germ pore; gelatinous cauda at each end. Two-celled ascospores uniseriate, biseriate, triseriate, or irregularly arranged; upper cell olivaceous brown, brown, or dark brown, ellipsoidal, ovoid, fusiform, navicular, or conical and triangular in lateral, smooth-walled, or spinulose to warty, with an apical or subapical pore, rarely 1-septate; lower cell short or long, hyaline to pale brown, occasionally brown, elongate fusiform, cylindrical, conical, triangular to hemispherical, rarely early evanescent, frequently collapsing, rarely up to 3-sepate; gelatinous appendages absent or present. Asexual morph absent or present. Conidiophores micronematous, reduced to conidiogenous cells. Conidiogenous cells intercalary or occasionally terminal, sometimes phialidic, originating lateral or terminal peg-like structure with a flaring collarette. Conidia blastic, single-celled, hyaline, globose to subglobose or pyriform, smooth-walled, usually with a truncated base and a rounded apex.

Notes: *Triangularia* included species traditionally characterized by ascospores with a conical, triangular in lateral view upper cell, and triangular or hemispherical lower cell [43,47]. Recently, this genus was restricted to the type species of the genus, *T. bambusae*, together with several species of *Apiosordaria* and *Podospora* that occurred in the same monophyletic clade [6]. All the species at that time included in *Triangularia* presented two-celled ascospores, although the lower cell of *T. allahabadensis* is soon evanescent. In our phylogenetic study, *Ar. arizonense* was nested in the *Triangularia* clade, and, therefore, a new combination is proposed here and, consequently, the description of the genus is emended to incorporate the possibility of finding one-celled ascospores. Moreover, *C. striata* and *Z. tetraspora*, both characterized by two-celled ascospores, were also located in the same clade with *Triangularia* spp., and new combinations are proposed here for these two taxa. Miller and Huhndorf [48] observed that *T. striata* produces a cladorrhinum-like asexual morph, which is one of the main characteristics that defines the recently described family Podosporaceae, where *Triangularia* belongs. See Notes of *Cladorrhinum* for morphological comparison with related genera.

***Triangularia arizonensis* (Griffiths) Y. Marín, A.N. Mill. & Stchigel, comb. nov**. MycoBank MB835504. (Figure 9A–F).

*Basionym: Pleurage arizonensis* Griffiths, Mem. Torrey Bot. Club 11: 57. 1901.

*Synonyms: Sordaria arizonensis* (Griffiths) Sacc. & D. Sacc., Syll. Fung. (Abellini) 17: 601. 1905.

*Podospora arizonensis* (Griffiths) Cain, Canad. J. Bot. 40: 459. 1962.

*Arnium arizonense* (Griffiths) N. Lundq. & J.C. Krug, Symb. Bot. Upsal. 20: 232. 1972.

***Triangularia striata* (Ellis & Everh.) Y. Marín, A.N. Mill. & Stchigel, comb. nov**. MycoBank MB835505. (Figure 10A–H).

*Basionym: Sordaria striata* Ellis & Everh., J. Mycol. 4: 79 [‘67’]. 1888.

*Synonyms: Lasiosordaria striata* (Ellis & Everh.) Chenant., Bull. Soc. Mycol. France 35: 78. 1919.

*Podospora striata* (Ellis & Everh.) Ellis & Everh., N. Amer. Pyren. (Newfield): 131. 1892.

*Cercophora striata* (Ellis & Everh.) N. Lundq., Symb. Bot. Upsal. 20: 105. 1972.

***Triangularia tetraspora* (J.N. Rai, Mukerji & J.P. Tewari) Y. Marín & Stchigel, comb. nov**. MycoBank MB835506.

*Basionym: Tripterospora tetraspora* J.N. Rai, Mukerji & J.P. Tewari, Canad. J. Bot. 41: 327. 1963.

*Synonyms: Zopfiella tetraspora* (J.N. Rai, Mukerji & J.P. Tewari) S. Ahmad, Monogr. Biol. Soc. Pakistan 7: 78. 1978.

*Podospora buffonii* Cailleux, Cah. Maboké 7: 12. 1969.

Key to species of *Triangularia*.
1. Only asexual morph produced*T. phialophoroides*1. Sexual morph produced22. Ascospores one-celled*T. arizonensis*2. Ascospores two-celled33. Asci 128-spored*T. setosa*3. Asci 4–8-spored44. Ascospores spinulose to warty*T. verruculosa*4. Ascospores smooth-walled under light microscope55. Lower cell early evanescent*T. allahabadensis*5. Lower cell not evanescent, frequently collapsing66. Lower cell short, conical, or triangular to hemispherical76. Lower cell long, elongate fusiform to cylindrical87. Lower cell conical, ascospores with mucilaginous small appendages at both ends*T. backusii*7. Lower cell triangular to hemispherical, ascospores without mucilaginous appendages*T. bambusae*8. Ascomata non-ostiolate98. Ascomata ostiolate109. Upper cell (11–)12–14.5(–15) × (7.5–)8–9(–9.5) µm*T. longicaudata*9. Upper cell 23.5–35.5 × 17.5–23.5 µm*T. tetraspora*10. Striate ascomata composed of areoles, upper cell up to 23 µm*T. striata*10. Above characters of the ascomata not present, upper cell up to 38 µm*T. anserina/comata/pauciseta* complex


Clade VIII (80% bs/1 pp; Figure 2) includes species of *Schizothecium*, *Echria*, *Immersiella*, *Jugulospora*, *Rinaldiella*, *Strattonia*, and *Zygopleurage*, plus several species of *Arnium*, *Apiosordaria*, *Cercophora*, *Triangularia*, and *Zopfiella*. This clade represents a new family, the Schizotheciaceae, which is proposed below.

**Schizotheciaceae Y. Marin & Stchigel, fam. nov**. MycoBank MB835507.

*Etymology*: Based on the genus *Schizothecium*, the oldest representative genus in the family.

*Type genus: Schizothecium* Corda, Icon. Fung. (Prague) 2: 29. 1838.

Ascomata ostiolate, immersed, semi-immersed, or superficial, scattered or aggregated, light to dark brown or black, globose to subglobose, pyriform, pyriform-conical, ovoid or obpyriform, glabrous or covered with flexous hairs, sometimes upper part of the ascomata adorned with groups of swollen agglutinated hairs, or with prominent protruding ascomatal cells; neck short or long, dark brown to black, opaque, conical, cylindrical, or elongate, papillate, sometimes with short, thick-walled, swollen cells, or swollen agglutinated hairs, or long, sometimes agglutinated, rigid hairs not forming fascicles, or with tufts of brown to black, rigid, agglutinated, septate, long, thick hairs; ascomatal wall pseudoparenchymatous, membranaceous, occasionally semi-coriaceous, semi-transparent, occasionally dark brown to black and opaque, yellowish to light brown, *textura angularis*, *textura globulosa*, or *textura epidermoidea*. Paraphyses and periphyses absent or present. Asci unitunicate, 4- to 2048-spored, clavate, cylindrical, broadly ellipsoid, or broadly fusiform, apically rounded, long or short stipitate, apical ring absent or present, dehiscing below the apex. Ascospores uniseriate, biseriate, or irregularly arranged, one- or two-celled. One-celled ascospores uniseriate or biseriate, hyaline, turning pale brown, golden, ochraceous, dark brown, or blackish brown at mature, cylindrical, fusiform or ellipsoidal, sigmoid or geniculate, one end may become swollen and turn brown, smooth-walled, without or with a germ pore at one or both ends, sometimes with a gelatinous appendage at each end, sometimes with a striated gelatinous sheath surrounding whole ascospore. Two-celled ascospores at first one-celled, hyaline, clavate, cylindrical or spatuliform, mostly becoming transversely uniseptate, occasionally developing two terminal ellipsoid black cells that remain connected with the long, often undulating, hyaline middle part of the spore, with a germ-pore at the free end and secondary appendages gelatinous arranged in rings around the top and base of the colored cells; upper cell olivaceous to brown to dark brown, globose, navicular, ellipsoidal, ellipsoidal-fusiform, fusiform, fusiform-obovoid, obovoid, or polygonal with five angles in side view, smooth-walled, finely granulated, pitted, ornamented with spines, or warted, sometimes with warts arranged forming ridges or large spots, with an apical or subapical germ pore; lower cell hyaline to pale brown, conical, cylindrical, cylindrical-obclavate, obclavate or obconical, collapsing or long persistent, smooth-walled to slightly warted; apical gelatinous appendages absent or present, attached at each end, lash-like; gelatinous sheaths rarely present, striated, swelling in water. Asexual morph absent or present. Conidiophores reduced to conidiogenous cells. Conidiogenous cells sometimes present, phialidic. Conidia hyaline to pale brown, ovoid to clavate or elongate, almost smooth-walled, usually produced laterally or terminally on undifferentiated hyphae, solitary.

Notes: This family is introduced to accommodate lasiosphaeriaceous taxa located in a well-supported clade (80% bs/1 pp; clade VIII, Figure 2), distinct from Lasiosphaeriaceae s. str. (clade V, Figure 1). It includes the genera *Echria*, *Immersiella*, *Jugulospora*, *Pseudoechria*, *Pseudoschizothecium*, *Rinaldiella*, *Rhypophila*, *Schizothecium*, and *Zygopleurage*, but also several species of *Apiosordaria* and *Zopfiella*. However, the type species of these latter genera are located in the Podosporaceae (clade IV) and in the Lasiopshaeriaceae s. str. (clade V), respectively. Consequently, those species that fall into clade VIII must be reviewed before they are moved to other (new) genera. Two species of *Arnium*, i.e., *A. cirriferum* and *A. hirtum*, are also included in this family. *Arnium* is a polyphyletic genus scattered throughout the order Sordariales, and its type species (*A. lanuginosum*) has not been sequenced. Finally, a strain identified as *Cercophora mirabilis* (which is the type species of the genus) was also located in the Schizotheciaceae, but its morphological study was not possible to confirm the identification, because the colonies remained sterile on different culture media. Due to this inconvenience, further studies are needed for the correct placement and delimitation of the genus *Cercophora*. The Schizotheciaceae are characterized by the production of mostly ornate, ostiolate ascomata with different kinds of structures: *Schizothecium* presents short swollen agglutinated hairs or prominent protruding ascomatal wall cells; *Echria* and *Pseudoechria* produce tufts of long rigid agglutinated hairs, while *Pseudoschizothecium* produces warts composed of small polygonal cells. The ascomatal wall is mostly pseudoparenchymatous, membranaceous, and semi-transparent. Ascospores display a large amount of variability including taxa with one- or two-celled ascospores, as well as ascospores with two independent cells that remain connected by a long, often undulating, hyaline middle part.

In the Schizotheciaceae, a well-supported clade (100% bs/1 pp; Figure 2) was composed of a reference strain of *Jugulospora rotula*, ATCC 38359; a reference strain of *Strattonia carbonaria*, ATCC 34567; the type strains of *Apiosordaria antarctica, Apiosordaria globosa, Apiosordaria hispanica*, and *Apiosordaria vestita,* CBS 381338, CBS 110113, CBS 110112, and IMI 381338, respectively; and an isolate morphologically identified by us as *Rhexosporium terrestre*, FMR 12428. All these taxa produce similar ascomata (with a translucent ascomatal wall and dark brown papillate neck), and clavate and early septate ascospores with the upper cell warted or finely granulate. Therefore, all these taxa are transferred to the genus *Jugulospora*, which is also emended.

***Jugulospora* N. Lundq., Symb. Bot. Upsal. 20(no. 1): 256. 1972, emend. Y. Marin & Stchigel**. MycoBank MB2538. (Figure 11A–N).

*Synonym: Rhexosporium* Udagawa & Furuya, Trans. Mycol. Soc. Japan 18: 302. 1977.

*Type species: Jugulospora rotula* (Cooke) N. Lundq.

Ascomata ostiolate, covered with pale brown, flexuous, septate, thick-walled hairs; neck dark brown to black, composed of papillate cells disposed around the ostiole; ascomatal wall membranaceous, semi-transparent, 3–9-layered, outer cells isodiametric. Paraphyses filiform. Asci unitunicate, eight-spored, cylindrical, with a thin apical ring. Ascospores at first one-celled, hyaline, clavate, early septate; upper cell brown, navicular, globose, or obovoid, warted, finely granulated, or pitted, sometimes with warts arranged forming ridges or large spots, with an apical germ pore; lower cell hyaline, conical to cylindrical, smooth-walled to slightly warted, collapsing; gelatinous sheath rarely present, hyaline, thin. Asexual morph absent or present. Conidia hyaline to pale brown, almost smooth-walled, ovate to elongate, produced laterally or terminally on undifferentiated hyphae, solitary.

Notes: The genus *Jugulospora* was introduced by Lundqvist [9] to accommodate *Sphaeria rotula* based on its verrucose ascospores with anastomosing warts arranged in large spots. The molecular data demonstrated that *A. globosa*, *A. hispanica*, and *R. terrestre*, also with warted ascospores, is synonymous with *J. rotula*. *Rhexosporium* was proposed as a new genus by Udagawa and Furuya [49] due to the presence of the longitudinal ridges in the upper cell of the ascospores, even though these authors cited a similar ascospore ornamentation in the genera *Apiosordaria* and *Jugulospora*. We noticed that such ascospore ridges seen under SEM (from the pictures taken by those authors) were really in a linear arrangement of warts and, consequently, not differing with respect to the ornamentation of those of *J. rotula*. Based on the early ascospore septation, the absence of staining of the ascal apical ring by cotton blue, and the presence of paraphyses, Lundqvist [9] highlighted the close similarity between *Jugulospora* and *Strattonia carbonaria*. Our study confirmed the phylogenetic relationship among them, already demonstrated in previous phylogenetic studies [11,26]. Therefore, we transferred *Strattonia carbonaria* to *Jugulospora* as *J. carbonaria*. Although *Strattonia* is an older name than Jugulospora, the type species of that genus, *S. tetraspora*, has an ascomatal neck with rigid, cylindrical, septate, long hairs and ascospores with a gelatinous sheath, features that do not match with those of Jugulospora. Hence, the genus *Strattonia* should be reduced to those species that share the morphology of the type species. Moreover, the type strains of *A. antarctica* and *A. vestita* clustered in the *Jugulospora* clade but formed independent lineages. Therefore, both species are transferred to *Jugulospora*.

***Jugulospora antarctica* (Stchigel & Guarro) Y. Marín & Stchigel, comb. nov**. MycoBank MB835663. (Figure 11N).

*Basionym: Apiosordaria antarctica* Stchigel & Guarro, in Stchigel, Guarro & Mac Cormack, Mycologia 95: 1219. 2003.

***Jugulospora carbonaria* (W. Phillips & Plowr.) Y. Marín, Stchigel, Guarro & A.N. Mill., comb. nov**. MycoBank MB812137. (Figure 11C,D,M).

*Basionym: Sphaeria carbonaria* W. Phillips & Plowr., Grevillea 2: 188. 1874.

*Synonyms: Podospora carbonaria* (W. Phillips & Plowr.) Niessl, Hedwigia 22: 156. 1883.

*Psilosphaeria carbonaria* (W. Phillips & Plowr.) Cooke & Plowr., Grevillea 7: 85. 1879.

*Sordaria carbonaria* (W. Phillips & Plowr.) Sacc., Syll. Fung. (Abellini) 1: 233. 1882.

*Strattonia carbonaria* (W. Phillips & Plowr.) N. Lundq., Symb. Bot. Upsal. 20: 269 (1972).

*Zopfiella carbonaria* (W. Phillips & Plowr.) Arx, Proc. Kon, Ned. Akad. Wetensch. C. 76: 291. 1973.

***Jugulospora rotula* (Cooke) N. Lundq., emend. Y. Marin, Dania García, Cano & Stchigel**. MycoBank MB315972. (Figure 11A,B,E–L).

*Basionym: Sphaeria rotula* Cooke, Handb. British Fungi 2: no. 2598. 1871.

*Synonyms: Rhexosporium terrestre* Udagawa & Furuya, Trans. Mycol. Soc. Japan 18: 303. 1977.

*Apiosordaria globosa* Dania García, Stchigel & Guarro, Mycologia 95: 137. 2003.

*Apiosordaria hispanica* Dania García, Stchigel & Guarro, Mycologia 95: 134. 2003.

Ascomata ostiolate, superficial or immersed, scattered to aggregated, pale brown to brown, pyriform, 350–770 × 200–540 µm, covered with pale brown, wide near the base, septate hyphae-like hairs of 1–5 µm diam; neck brown to dark brown, cylindrical to conical, papillate, 78–280 µm long, 90–250 µm wide; ascomatal wall membranaceous, semi-transparent, brownish-orange to brown, 3–9-layered, 15–45 µm thick; outer layers with *textura angularis* to *textura intricata*; inner layers with *textura epidermoidea*. Paraphyses and periphyses filiform, up to 2 µm in diam. Asci unitunicate, eight-spored, cylindrical, 146–250 × 14–28 µm, stipitate, with a thin apical ring, evanescent. Ascospores at first one-celled, hyaline, clavate, becoming transversely septate and two celled; upper cell dark brown, obovoid to globose, truncate at the base, ornamented with warts arranged uniformly or forming longitudinal ridges or large spots, 18–29 × 12–27 µm, with an apical to lateral germ pore of 0.5–3 µm; lower cell hyaline, conical, smooth-walled to slightly warted, 1–6 µm, collapsing; gelatinous cauda absent. Asexual morph absent or present. Conidia hyaline to pale-colored, almost smooth-walled, ovate to elongate, 2–6 × 1.5–2.5 µm, produced laterally or terminally on undifferentiated hyphae, solitary.

*Specimens examined*: USA, NORTH CAROLINA, Great Smoky Mountains National Park, from soil, 8-VIII-2008, M. Calduch, A.N. Miller & A.M. Stchigel, culture FMR 12690; USA, TENNESSEE, Great Smoky Mountains National Park, from soil, 9-VIII-2008, M. Calduch, A.N. Miller & A.M. Stchigel, culture FMR 12781; SPAIN, TARRAGONA, Els Gorgs de la Febró, from soil, 25-IX-1996, A.M. Stchigel & M. Calduch, culture CBS 110112, ex-type strain of *Apiosordaria hispanica*; SPAIN, TARRAGONA, Els Gorgs de la Febró, from soil, 25-IX-1996, A.M. Stchigel & M. Calduch, culture CBS 110113, ex-type strain of *Apiosordaria globosa*; ANTARCTICA, King George Island, Jubany Argentinian base, from soil, 11-XI-1996, W. Mac Cormack, culture IMI 381338, ex-type strain of *Apiosordaria antarctica*; JAPAN, from burned soil, Y. Horie, culture ATCC 34567.

***Jugulospora vestita* (Udagawa & Y. Horie) Y. Marín & Stchigel, comb. nov**. MycoBank MB835664.

*Basionym: Apiosordaria vestita* Udagawa & Y. Horie, Reports on the Cryptogamic Study in Nepal (Tokyo): 97. 1982.

Key to species of *Jugulospora*.
1. Ascospores pitted, with gelatinous sheath*J. vestita*1. Ascospores warty or finely granulated, without gelatinous sheath22. Ascospores warty*J. rotula*2. Ascospores finely granulated33. Ascomata with short neck*J. carbonaria*3. Ascomata with long neck, up to 400 μm*J. antarctica*

The type strain of *T. tanzaniensis* and a reference strain of *Z. karachiensis,* which are characterized by the formation of ostiolate ascomata with membranaceous to semi-coriaceous ascomatal walls, and ascospores with a smooth-walled upper cell, were included in a well-supported clade (100% bs/1 pp; Figure 2). Consequently, we propose the erection of the new genus *Lundqvistomyces* as follows:

***Lundqvistomyces* Y. Marín & Stchigel, gen. nov**. MycoBank MB835508.

*Etymology*: Named in honor of the mycologist Nils Lundqvist, who contributed greatly to the taxonomy of Lasiosphaeriaceae.

*Type species: Lundqvistomyces karachiensis* (S.I. Ahmed & Asad) Y. Marín & Stchigel.

Ascomata ostiolate, immersed to almost superficial, scattered or aggregated, brown to nearly black, pyriform, covered with brown, flexous hairs; neck black, conical to papilliform; ascomatal wall membranaceous to semi-coriaceous, dark brown to black, opaque, *textura angularis*. Periphyses absent or present, hyaline, filiform. Paraphyses hyaline, filiform. Asci unitunicate, eight-spored, clavate or cylindrical-clavate, thin-walled, short stipitate, with an indistinct apical ring. Ascospores biseriate, at first one-celled, ellipsoidal, becoming transversely uniseptate and two-celled; upper cell dark, ellipsoidal-fusiform to fusiform, sometimes slightly inequilateral, with a truncate base and an apical germ pore; lower cell hyaline to pale brown, conical, often collapsing; gelatinous appendages absent. Asexual morph absent or present. Conidia absent or delimited from peg-like non-proliferating structures on undifferentiated hyphae.

Notes: This genus includes two species previously identified as *T. tanzaniensis* and *Z. karachiensis*. In our phylogenetic study, both species were located in an independent terminal clade (100% bs/1 pp) in the Schizotheciaceae (clade VIII), far from the type species of *Triangularia*, located in the Podosporaceae (clade IV), and from *Zopfiella*, in the Lasiosphaeriaceae s. str. (clade V). *Lundqvistomyces* is characterized by its ostiolate ascomata with a membranaceous to semi-coriaceous ascomatal wall of *textura angularis*, asci with an indistinct apical ring, and ascospores at the first ellipsoidal, and with a smooth-walled upper cell. *Lundqvistomyces karachiensis* differs from *L. tanzaniensis* mainly in the size of the upper and lower cells of the ascospores (broader upper cell and longer lower cell in the latter species). Moreover, *L. karachiensis* also produces an asexual morph, which is absent in *L. tanzaniensis*.

***Lundqvistomyces karachiensis* (S.I. Ahmed & Asad) Y. Marín & Stchigel, comb. nov**. MycoBank MB835509.

*Basionym: Strattonia karachiensis* S.I. Ahmed & Asad, Sydowia 21: 282. 1968.

*Synonyms: Zopfiella karachiensis* (S.I. Ahmed & Asad) Guarro, in Guarro & Cano, Trans. Brit. Mycol. Soc. 91: 589. 1988.

*Triangularia karachiensis* (S.I. Ahmed & Asad) Udagawa, in Udagawa, Tsuzaki & Uehara, Trans. Mycol. Soc. Japan 20: 362. 1979.

*Podospora faurelii* Mouch., Rev. Mycol. (Paris) 38: 109 and 112. 1975.

***Lundqvistomyces tanzaniensis* (R.S. Khan & J.C. Krug) Y. Marín & Stchigel, comb. nov**. MycoBank MB835510.

*Basionym: Triangularia tanzaniensis* R.S. Khan & J.C. Krug, Mycologia 81: 865. 1990.

*Synonym: Zopfiella tanzaniensis* (R.S. Khan & J.C. Krug) Guarro, P.F. Cannon & Aa, Syst. Ascomycetum, Reprint of Volumes 1-4 (1982-1985) 10: 103. 1991.

Reference strains of *Podospora curvicolla, P. decidua, P. longicollis*, and *P. prolifica*, grouped in a well-supported independent clade (100% bs/1 pp; Figure 2) within the family Schizothecaceae. These taxa are characterized by ascomata with tufts of long agglutinated hairs on the neck and asci usually containing more than eight ascospores. Consequently, we introduce the new genus *Pseudoechria* as follows.

***Pseudoechria* Y. Marín & Stchigel, gen. nov**. MycoBank MB835511. (Figure 12A–I).

*Etymology:* Based on the morphological resemblance with *Echria*, which also presents tufts of rigid agglutinated long hairs on the neck of the ascomata.

*Type species: Pseudoechria curvicolla* (G. Winter) Y. Marín & Stchigel.

Ascomata ostiolate, immersed or superficial, scattered or aggregated, ovoid, broadly obpyriform or globose, sometimes with flexuous, light brown septate, thick-walled hairs; neck short or long, dark, opaque, with tufts of brown to black, rigid, agglutinated, septate, long, thick-walled hairs; ascomatal wall membranaceous, translucent, light olive brown to black, or yellowish to light brown with an olivaceous tinge, thin, of *textura angularis*. Paraphyses absent or present, more or less thread-like, soon evanescent. Asci 8- to 2048-spored, clavate to broadly ellipsoid, or broadly fusiform, short-stipitate, with or without a thick apical ring. Ascospores multiseriate, at first one-celled, hyaline, becoming transversely uniseptate; upper cell olivaceous to dark brown, ellipsoidal, sometimes with subacute ends, with an apical to subapical germ pore; lower cell hyaline, clavate, soon evanescent; gelatinous cauda attached at each end, lash-like. Asexual morph absent.

Notes: *Pseudoechria* is introduced to accommodate four species of *Podospora*, which form a monophyletic clade in the Schizotheciaceae (clade VIII), and were distantly related to *Podospora* s. str., located in Podosporaceae (clade IV). This genus is characterized by ascomata with tufts of long agglutinated hairs on the neck and (mostly) more than eight-spored per asci. A similar morphology is observed in *Schizothecium*, which also has ascomata with ornations at the apex and multi-spored asci. In fact, *Pse. longicollis* was placed in *Schizothecium* when it was introduced [50]. Both genera differ in the type of ascomata ornamentation, with groups of swollen, agglutinated, short hairs or prominent protruding wall cells in *Schizothecium*, while in *Pseudoechria*, the vestiture comprises tufts of rigid, agglutinated, long hairs. Rigid long hairs can also be present in *Echria* spp. and *Arnium hirtum*, which are also located in the Schizotheciaceae. Both taxa can be easily distinguished from Pseudoechria by their one-celled ascospores. Moreover, the hairs in *A. hirtum* are not grouped in tufts as in the other two genera, and this species presents also agglutinated hairs as in *Schizothecium*, which is not observed in either *Echria* or *Pseudoechria*. *Pseudoschizothecium*, located in Schizotheciaceae, and Rhypophila, located in Naviculisporaceae, also present similar morphology with an ornated ascomata apex and multi-spored asci. *Rhypophila* differs in having tuberculate projections, whereas Pseudoschizothecium produces warts composed of smaller polygonal cells. Although *Triangularia setosa* also presents multi-spored asci and ascomata with long and rigid hairs on the neck, as in the genus *Pseudoechria*, the ascomatal hairs are mostly solitary. However, our phylogenetic analysis confirmed that this species, which was located in the family Podosporaceae (clade IV), is not related to *Pseudoechria*.

Two strains of *Pse. curvicolla* included in the phylogenetic study grouped in a monophyletic clade (100% bs/0.99 pp) together with the ex-type strain of *Pse. longicollis*. However, only ITS and LSU sequences of *Pse. curvicolla* were available; unfortunately, these two loci display less variability. Consequently, *rpb2* and *tub2* sequences are necessary to confirm the potential synonymy of both species.

***Pseudoechria curvicolla* (G. Winter) Y. Marín, A.N. Mill. & Stchigel, comb. nov**. MycoBank MB835512. (Figure 12A–C,F,G)

*Basionym: Sordaria curvicolla* G. Winter, Abh. Naturf. Ges. Halle 13: 34. 1871.

*Synonyms: Podospora curvicolla* (G. Winter) Niessl, Hedwigia 22: 156. 1883.

Other synonyms in [9].

***Pseudoechria longicollis* (L.M. Ames) Y. Marín & Stchigel, comb. nov**. MycoBank MB835662.

*Basionym: Schizothecium* longicolle L.M. Ames, Sydowia 5: 121. 1951.

*Synonyms: Pleurage longicollis* (L.M. Ames) Boedijn, Persoonia 2: 312. 1962.

*Podospora longicollis* (L.M. Ames) J.H. Mirza & Cain, Canad. J. Bot. 47: 2028. 1970. (1969).

***Pseudoechria decidua* (Cailleux) Y. Marín, A.N. Mill. & Stchigel, comb. nov**. MycoBank MB835513. (Figure 12D,H)

*Basionym: Podospora decidua* Cailleux, Cah. Maboké 7: 102. 1969.

***Pseudoechria prolifica* (Cailleux) Y. Marín, A.N. Mill. & Stchigel, comb. nov**. MycoBank MB835514. (Figure 12E,I)

*Basionym*: Podospora prolifica Cailleux, Cah. Maboké 7: 102. 1969.

Key to species of *Pseudoechria*.
1. Asci 8-spored*Pse. decidua*1. Asci more than 8-spored22. Asci 1024-2048-spored, ascospores with upper cell up to 14 μm and lower cell up to 4 μm*Pse. prolifica*2. Asci 256- or 512-spored, ascospores with upper and lower cell longer than 14 and 4 μm, respectively33. Asci about 256-spored, ascospores with upper cell up to 16 μm long and lower cell up to 9 μm*Pse. curvicolla*3. Asci about 512-spored, ascospores with upper cell up to 22 μm and lower cell up to 8 μm *Pse. longicollis*

The genus *Pseudoschizothecium* is introduced to accommodate *C. atropurpurea*, which represents a basal independent lineage within the Schizotheciaceae (clade VIII), and that can be morphologically distinguished by the other members of the family by its dark purple warty ascomata and ascospores up to 5-septate.

***Pseudoschizothecium* Y. Marín, A.N. Mill. & Stchigel, gen. nov**. MycoBank MB835515. (Figure 13A–N).

*Etymology*: Based on the morphological resemblance to the genus Schizothecium.

*Type species: Pseudoschizothecium atropurpureum* (A.N. Mill. & Huhndorf) Y. Marín, A.N. Mill. & Stchigel.

Ascomata ostiolate, superficial, scattered or aggregated, dark purple to black, ovoid, papillate, warty toward the apex; neck conical, warty; warts small, hyaline to brownish, composed of polygonal cells; ascomatal wall pseudoparenchymatous, *textura angularis*. Periphyses present, filiform, hyaline, up to 2 µm in diam. Paraphyses filiform to venticrose, septate, soon evanescent. Asci unitunicate, eight-spored, elongate-clavate, long-stipitate, with a narrow and refractive apical ring, subapical globule large. Ascospores biseriate to triseriate into the asci, at first one-celled, hyaline, cylindrical, slightly sigmoid or geniculate, becoming transversely uniseptate; upper cell brown to dark brown, narrowly fusiform to ellipsoidal, occasionally 1-septate; lower cell hyaline to pale brown, sometimes brown, cylindrical, long, up to 3-septate. Asexual morph absent or present. Conidiophores reduced to conidiogenous cells. Conidiogenous cells phialidic, flask-shaped or elongate-clavate, some with distinct collarettes. Conidia hyaline, globose or ovoid.

Notes: *Pseudoschizothecium* is introduced to accommodate a single species, *C. atropurpurea*. This genus differs from all taxa included in the family by its dark purple warty ascomata and up to 5-septate ascospores (1-septate upper cell, and up to 3-septate lower cell). The same pattern of septation can be observed in *T. striata*, but this species can be distinguished by its areolate and striate ascomata. Ascospores with a septate lower cell are also seen in *Ar. ambigua*, which also has a septate upper cell. However, *Ar. ambigua* can be easily distinguished by its areolate cephalothecoid ascomatal wall. Our phylogenetic study demonstrated that these three taxa are not related, grouping in different families, i.e., *Areotheca ambigua* in Naviculisporaceae, *Pseudoschizothecium atropurpureum* in Schizotheciaceae, *Triangularia striata* in Podosporaceae. See Notes of *Pseudoechria* for morphological comparison with other genera with ornamented ascomata.

***Pseudoschizothecium atropurpureum* (A.N. Mill. & Huhndorf) Y. Marín, A.N. Mill. & Stchigel, comb. nov**. MycoBank MB835516. (Figure 13A–N).

*Basionym: Cercophora atropurpurea* A.N. Mill. & Huhndorf, Sydowia 53: 212. 2001.

***Schizothecium* Corda, Icon. Fungorum 2: 29. 1838. emend. Y. Marin & Stchigel**. MycoBank MB4908. 

*Type species: Schizothecium fimicola* Corda.

Ascomata ostiolate, immersed, semi-immersed or superficial, scattered or gregarious, light to dark brown, globose to subglobose, pyriform, or pyriform-conical, translucent, upper part ornate with groups of swollen agglutinated hairs, or with prominent protruding ascomatal wall cells; ascomatal wall membranaceous, pseudo-parenchymatous, *textura angularis* to *textura globulosa*. Paraphyses usually absent. Asci unitunicate, four-, eight-, or multi-spored, clavate to cylindrical, apically rounded, short- or long-stipitate, usually lacking apical structures, dehiscing below the apex. Ascospores at first one-celled, hyaline, becoming transversely uniseptate and two-celled; upper cell finally dark brown to blackish-brown, fusiform-obovoid, rarely heart-shaped to lunate and strongly curved and concave on one side, smooth-walled, with an upper germ pore; lower cell hyaline, rarely subhyaline, cylindrical, obclavate or obconical, usually long persistent; gelatinous appendages absent or present. Asexual morph absent or present. Conidiophores reduced to conidiogenous cells. Conidiogenous cells phialidic, flask-shaped or elongate-clavate, some with distinct collarettes. Conidia hyaline, globose or ovoid.

Notes: *Schizothecium* was morphologically and phylogenetically studied by Cai et al. [4]. As a result, the genus was delimited to 24 species, all them with ostiolate ascomata ornate with swollen agglutinated hairs or prominent protruding ascomatal wall cells. In our phylogenetic study, *P. selenospora*, which produces similar ascomata, grouped in this monophyletic clade. Therefore, the new combination *S. selenosporum* is proposed here, and the description of *Schizothecium* is emended to incorporate its ascospore morphology. This species is distinguished from all other species of *Schizothecium* by the heart-shaped to lunate shape of the ascospore upper cell. A key to distinguish all species of Schizothecium is available in Cai et al. [4].

Because the type strain of *P. selenospora* falls within the *Schizothecium* clade (100% bs/1 pp; Figure 2), the new combination for this taxon is proposed here.

***Schizothecium selenosporum* (Stchigel, Guarro & M. Calduch) Y. Marín & Stchigel, comb. nov**. MycoBank MB835517.

*Basionym: Podospora selenospora* Stchigel, Guarro & M. Calduch, Mycologia 94: 554. 2002.

## 4. Discussion

We performed a phylogenetic study based on the analysis of the ITS, LSU, *rpb2*, and *tub2* sequences of members of the Sordariales to contribute to a more natural classification of this order. Some families and genera included in our study have been previously reported to be polyphyletic [6,10,11,26,33]. Recently, extensive phylogenetic studies have been performed to properly delimit the family Chaetomiaceae, and its largest genera, *Chaetomium* and *Thielavia* [5,6,7]. In the revision of *Chaetomium*, five new genera (i.e., *Amesia*, *Arcopilus*, *Collariella*, *Dichotomopilus*, and *Ovatospora*) were introduced to accommodate species not falling into the monophyletic clade including the type species of the genus, *Ch. globosum* [5]. *Thielavia* was also recently revised, restricting the genus to the type species, *Th. basicola*, which occurred in the order Melanosporales. The remaining species of *Thielavia* grouped in the Sordariales and were transferred to new genera in the Chaetomiaceae, i.e., *Carteria, Chrysanthotrichum, Condenascus, Hyalosphaerella, Microthielavia, Parathielavia, Pseudothielavia, Stolonocarpus,* and *Thermothielavioides* [6].

The family Sordariaceae was considered monophyletic [2]; however, the genus *Diplogelasinospora*, previously belonging in this family, was placed closer to the Lasiosphaeriaceae [4,6]. In our study, the family Diplogelasinosporaceae has been introduced to accommodate *Diplogelasinospora*, which occurred as an independent lineage in the Sordariales. This new family is characterized by the production of arthroconida and asci lacking apical structures, which is also observed in Chaetomiaceae and Sordariaceae. However, these latter two families include only taxa producing one-celled ascospores, while ascospores are two-celled in Diplogelasinosporaceae.

Lasiosphaeriaceae has been considered a polyphyletic family [2,11]. Kruys et al. [26] previously separated the family into four groups (I, II, III, and IV) according to the clades obtained in their phylogenetic analysis based on LSU and *tub2*. Recently, the new family Podosporaceae was introduced to accommodate taxa included in their monophyletic group IV [6], which is phylogenetically distant from the monophyletic clade including the type genus of Lasiosphaeriaceae, *Lasiosphaeria*, together with other taxa including the type species of *Zopfiella*, *Z. tabulata*. In the present work, the family Lasiosphaeriaceae s. str. is reduced to the taxa included in this latter monophyletic clade, and the families Naviculisporaceae and Schizotheciaceae are introduced to accommodate lasiosphaeriaceous genera included in two other well-supported lineages. Lasiosphaeriaceae s. str. includes taxa with mostly ostiolate ascomata covered with a tomentum or hyphae-liked hairs below the neck. Naviculisporaceae is characterized by producing mostly ostiolate ascomata with membranaceous and pseudoparenchymatous ascomatal walls. Schizotheciaceae includes taxa that produce ostiolate ascomata with walls mostly pseudoparenchymatous, membranaceous, and semi-transparent, and frequently adorned with different kinds of ornamentation on the neck, or sometimes on the entire ascomata. Even though the ascomatal morphology is similar among members of the Naviculisporaceae, they show a large variability in ascospore morphology, including both one-celled and two-celled ascospores. *Arnium caballinum*, *A. japonense*, and *A. mendax* produce one-celled ascospores [9,39], while all other species produce two-celled ascospores. The presence of taxa with both kinds of ascospores is also observed in Lasiosphaeriaceae and Schizotheciaceae. In Naviculisporaceae, the ascomata are frequently ostiolate, except for *Z. marina*, *Z. pilifera*, and *Z. submersa*, which are non-ostiolate [40,41,42]. In our phylogenetic study, these three species were located in a well-supported clade together with *T. mangenotii*, which produces ostiolate ascomata [43], suggesting that these could represent a new genus; further phenotypic and molecular studies are necessary to demonstrate this assumption. Regarding the nature of the ascomatal wall in members of Naviculisporaceae, the membranaceous type is the most common. However, the new genus *Areotheca*, located in this family, is characterized by a carbonaceous areolate cephalothecoid ascomatal wall.

Recent molecular studies have demonstrated that the traditional circumscriptions of most of the genera included in the Lasiosphaeriaceae are artificial, as the ascospores morphology is an extremely homoplastic character not useful in predicting phylogenetic relationships [10,11]. Moreover, the morphology of the ascospores is not always useful as a taxonomic criterion to separate genera as was reported in the Sordariales, e.g., the genus *Gelasinospora* was synonymized with *Neurospora* despite the different patterns of ascospore ornamentation [51,52]. On the other hand, Miller and Huhndorf [11] demonstrated that in the Sordariales, the structure of the ascomatal wall is clearly more useful for delimitation of some genera. An example is *Schizothecium*, which was delimited for those species producing ostiolate ascomata with swollen agglutinated hairs or prominent protruding ascomatal wall cells [33]. Unfortunately, the type of ascomatal wall varies in taxa in the same phylogenetic clade [11], indicating that this character is not always useful for genera delimitation. Moreover, a large number of lasiosphaeriaceous fungi possess relatively simple ascomatal walls without discriminative characters [26].

Recently, several of the largest genera included in the Sordariales, i.e., *Apiosordaria*, *Podospora*, and *Triangularia*, were studied and delimited based on ITS, LSU, *rpb2*, and *tub2* sequence data. As a result, all three genera together with *Cladorrhinum* were placed into the family Podosporaceae. *Podospora* was restricted to the type species, *P. fimicola*, and *P. bulbillosa*, which was a new combination from a species formerly placed in *Cladorrhinum*. In the present study, *Ap. sacchari*, *Ap. striatispora*, and *C. costaricensis* were transferred to *Podospora* based on sequence data. *Apiosordaria* has been synonymized with *Triangularia*, as both type species, *A. verruculosa* and *T. bambusae*, clustered together. *Triangularia* was limited to species with two-celled ascospores [6]. In our phylogenetic study, *Ar. arizonense*, which forms one-celled ascospores, was nested in the *Triangularia* clade so the description of *Triangularia* was emended. *Cercophora striata* and *Z. tetrapora* were also located in the *Triangularia* clade, and new combinations were proposed. The genus *Cladorrhinum* (Podosporaceae) was characterized by taxa producing one-celled ascospores [6]. However, our phylogenetic study demonstrated that *C. coprophila*, which presents two-celled ascospores, also belongs to this genus. Additionally, *Ar. tomentosum* was also transferred to *Cladorrhinum*, and the synonymy between *Ar. olerum* and *Cl. foecundissimum* was proposed. Nevertheless, there are still a large number of species of *Apiosordaria*, *Podospora*, and *Triangularia* that need to be more extensively studied to determine their correct taxonomic placement. We demonstrated that four species of *Podospora*, i.e., *P. cochleariformis, P. decipiens, P. myriaspora,* and *P. pleiospora*, were located far from the type species of the genus, even in another family (Naviculisporaceae), and was consequently transferred to the new genus, *Rhypophila*. This genus is characterized by the production of ascomata with a neck bearing elongated tuberculated projections at the base, asci mostly containing more than eight ascospores, and two-celled ascospores in which the lower cell is as long as, or longer than, the upper cell. Moreover, two new genera, *Pseudoechria* and *Pseudoschizothecium*, were introduced to accommodate five other species of *Podospora* included in two different monophyletic clades in the new family Schizotheciaceae. *Pseudoechria* species possess tufts of rigid agglutinated long hairs adorning the neck of the ascomata, while *Pseudoschizothecium* produces warts composed of smaller polygonal cells. Moreover, *Pseudoschizothecium* can be easily distinguished by its up to 5-septate ascospores. We also introduced the new genus *Lundqvistomyces* to accommodate *T. tanzaniensis* and *Z. karachiensis*. This genus produces ostiolate ascomata with a membranaceous to semi-coriaceous ascomatal wall of *textura angularis*, asci with an indistinct apical ring, and ascospores at the first ellipsoidal, and with an upper cell smooth-walled when mature. Finally, four species of *Apiosordaria*, i.e., *Ap. antarctica, Ap. globosa, Ap. hispanica* and *Ap. vestita*, were transferred to *Jugulospora*, which is also placed in the Schizotheciaceae. This genus is characterized by ostiolate ascomata covered with flexous hairs and a nearly black papillate neck, a membranaceous and semi-translucent ascomata wall, asci with an apical ring, and ascospores with early septation. The ornamentation of the ascospores is variable, from finely granulated to warted. Therefore, it is another clear example that the ascomatal wall is a better predictor of phylogeny compared to the ascospore morphology, as has been previously pointed out [10].

The delimitation of some lasiosphaeriaceous genera remains confusing, e.g., *Arnium*, *Cercophora*, and *Zopfiella*. *Arnium* comprises species with different kinds of ascomata, from membranaceous and pale-colored to coriaceous or carbonaceous and opaque, and with one-celled dark ascospores that are sometimes 1-septate and often with gelatinous appendages or sheaths [9,26]. Phylogenetic studies demonstrated that this genus is polyphyletic and its species are scattered throughout the order Sordariales [26], in the families Naviculisporaceae, Podosporaceae, and Schizotheciaceae. The main problem surrounding the proper placement of *Arnium* is the lack of sequence data for the type species of the genus, *Ar. lanuginosum*.

*Cercophora* is the largest polymorphic genus in the Lasiosphaeriaceae s.l. and includes approx. 80 species [9]. It is characterized by ostiolate ascomata with a pseudoparenchymatous ascomatal wall and two-celled ascospores, usually with gelatinous cauda at both ends [9]. The type species, *C. mirabilis*, occurred on an independent branch in the Schizotheciaceae. However, the strain included is not from the type material and its morphological study was not possible, because the colonies remained sterile in different culture media. Therefore, we decided not to reduce the genus to this single species until further studies can be performed including the examination of type material or reference strains whose morphology can be studied. The genus *Areotheca* was introduced to accommodate *C. areolata* and *C. ambigua*, which produce ascomata with a carbonaceous, cephalothecoid, areolate ascomatal wall that cracks in polyhedral plates. The presence of areoles is shared by *T. striata*, a few species of *Cladorrhinum*, i.e., *Cl. coprophilum* and *Cl. tomentosum*, and a few other *Cercophora* species, i.e., *C. caerulea*, *C. septentrionalis,* and *C. silvatica*. However, the areoles are not fully developed or carbonaceous in any of these species as compared to *Areotheca*. Further studies need to be performed to correctly place all the species of *Cercophora* with this particular ascomatal wall. *Cercophora scortea*, occurred in an independent lineage in the Lasiosphaeriaceae s.l., together with *P. appendiculata* and these taxa probably represent a new genus. Both species are characterized by a coriaceous and pseudo-bombardioid ascomatal wall [53].

The genus *Zopfiella*, introduced by Winter [54], is characterized by usually non-ostiolate, membranaceous to coriaceous ascomata, more or less clavate asci, and two-celled ascospores with a dark and smooth-walled upper cell, and a hyaline or pale brown lower cell [55]. As the type strain of *Zopfiella*, *Z. tabulata*, possessed a septate upper cell, Cain [56] established *Tripterospora* to accommodate species formerly in *Zopfiella* that lacked a septate upper cell. However, in 1971, Malloch and Cain [57] redefined both genera, distinguishing each genus by the shape of the immature ascospores, which is ellipsoidal or cylindrical in *Zopfiella* and clavate in *Tripterospora*, and by the lower cell, broad and persistent in *Zopfiella*, and narrow and collapsing in *Tripterospora*. Moreover, they did not consider the presence or absence of a septum in the upper cell as a generic character. In our phylogenetic study, *Z. tabulata* grouped in a monophyletic clade together with *Anopodium ampullaceum, Bellojisia rhynchostoma, C. sparsa, C. sulphurella, Corylomyces selenospora,* and *P. didyma*. All of these species (except *An. ampullaceum*) are characterized by a septate upper cell. Therefore, septation is a good character to delimitate this genus, as was previously suggested by Cai et al. [46]. However, because the strain of *Z. tabulata* included in that work is not the ex-type strain, and because it was not morphologically examined in our study, the delimitation of this genus was not performed until further studies can be conducted. The new monotypic genus, *Naviculispora*, isolated from soil, is also characterized by the presence of a septum in the upper cell, but it was placed in a new family, the Naviculisporaceae, while *Z. tabulata* belongs to the Lasiosphaeriaceae s. str. *Naviculispora* can be easily distinguished from taxa with a septate upper cell included in the same clade than *Z. tabulata* by its brownish, thick-walled, persistent lower cell. *Areotheca* (Naviculisporaceae) also produces ascospores with a septate upper cell, but this genus can be easily distinguished by its areolate ascomatal wall.

Finally, *Bombardia bombarda* and *Bombardioidea anartia* constituted a well-supported clade in the Lasiosphaeriaceae s. l., with a nucleotide similarity of 98% among the two taxa. Both genera share a particular ascomal wall, i.e., bombardioid wall composed of two stromatic outer layers that is unique in the order. This suggests that both taxa could be placed in the same genus and a new family created in the Sordariales, but further studies including additional species and the inclusion of type material are needed to confirm these hypotheses. Other lineages, such as the one including the genera *Lasiosphaeris* and *Zygospermella*, and another involving the species of *Apodospora*, may represent new families in the Sordariales, but further studies including more strains are necessary to verify these ideas.

## Figures and Tables

**Figure 1 microorganisms-08-01430-f001:**
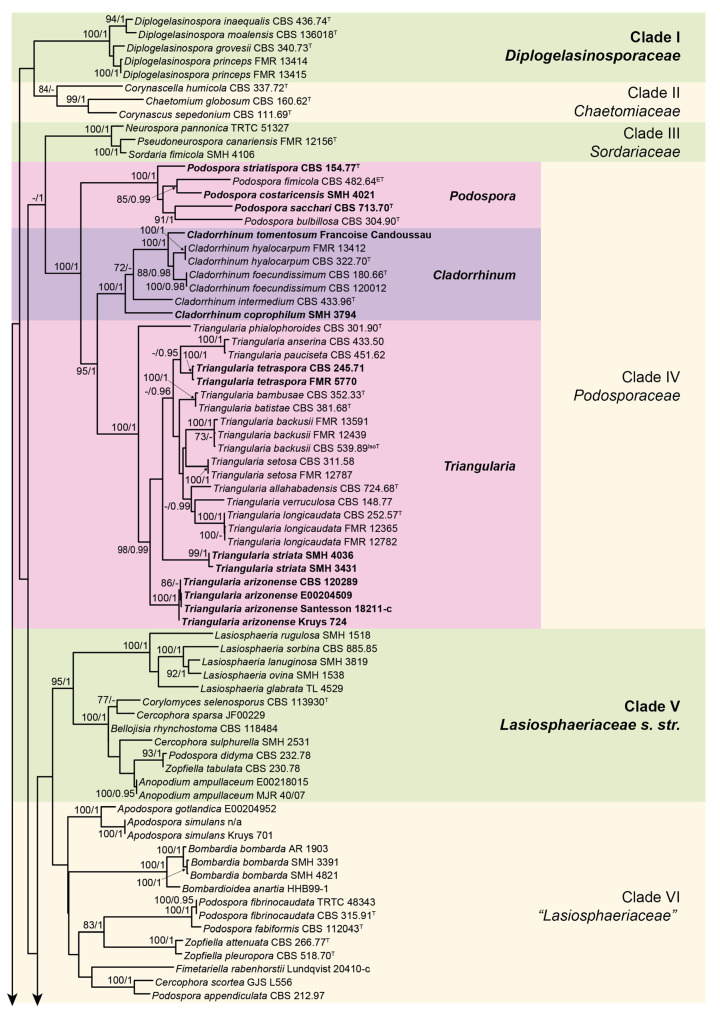
RAxML phylogram obtained from the combined internal transcribed spacer region (ITS), the nuclear rDNA large subunit (LSU), and fragments of ribosomal polymerase II subunit 2 (*rpb2*) and β-tubulin (*tub2*) genes sequences of our isolates and selected strains belonging to the families Chaetomiaceae, Diplogelasinosporaceae, Lasiosphaeriaceae, Naviculisporaceae, Podosporaceae, Schizotheciaceae, and Sordariaceae. *Camarops amorpha* SMH 1450 was used as an outgroup. Bootstrap support values ≥70/Bayesian posterior probability scores ≥0.95 are indicated along branches. Branch lengths are proportional to distance. Novelties and emended taxa are indicated in **bold**. Ex-epitype, ex-isotype, and ex-type strains of the different species are indicated with ^ET^, ^IsoT^, and ^T^, respectively.

**Figure 2 microorganisms-08-01430-f002:**
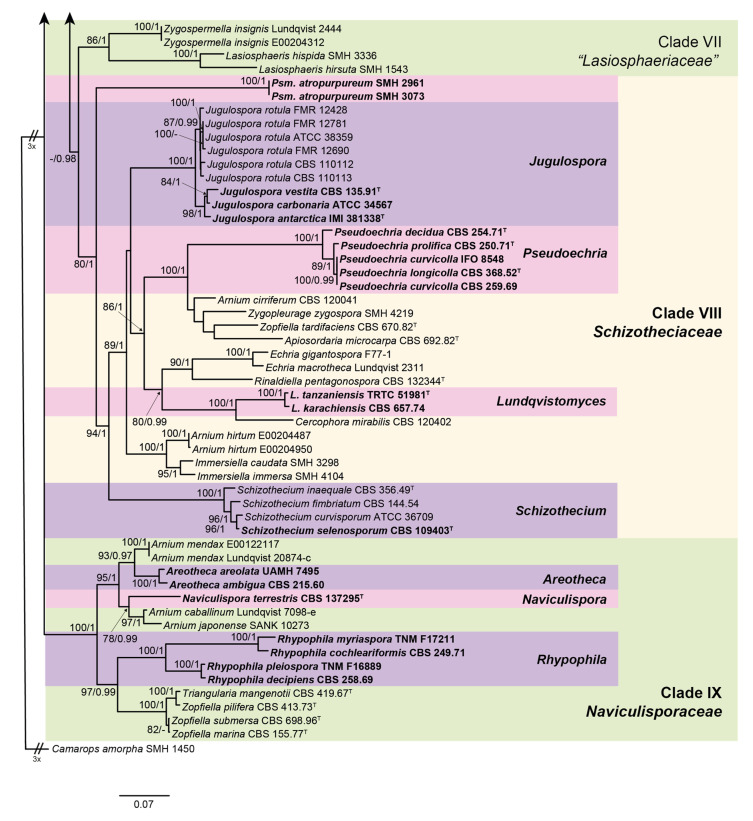
Second part of the RAxML phylogram obtained from the combined ITS, LSU, *rpb2*, and *tub2* sequences of our isolates and selected strains belonging to the families Chaetomiaceae, Diplogelasinosporaceae, Lasiosphaeriaceae, Naviculisporaceae, Podosporaceae, Schizotheciaceae, and Sordariaceae. *Camarops amorpha* was used as an outgroup. Bootstrap support values ≥70/Bayesian posterior probability scores ≥0.95 are indicated along branches. Branch lengths are proportional to distance. Novelties and emended taxa are indicated in **bold**. Ex-epitype, ex-isotype, and ex-type strains of the different species are indicated with ^ET^, ^IsoT^, and ^T^, respectively.

**Figure 3 microorganisms-08-01430-f003:**
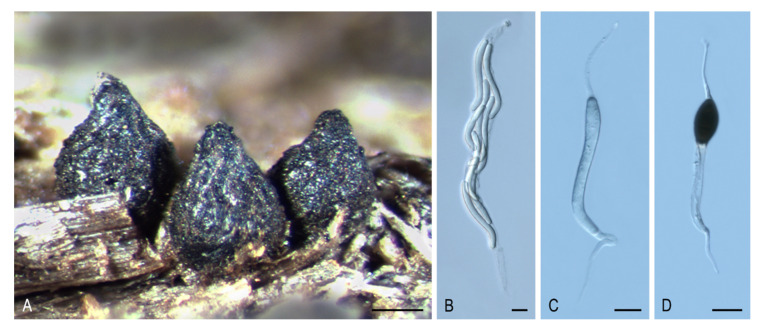
*Areotheca areolata* JF05120_ANMAcc49. (**A**) Ascomata. (**B**) Ascus. (**C**,**D**) Ascospores. Bars: (**A**) = 200 μm; (**B**–**D**) = 10 μm.

**Figure 4 microorganisms-08-01430-f004:**
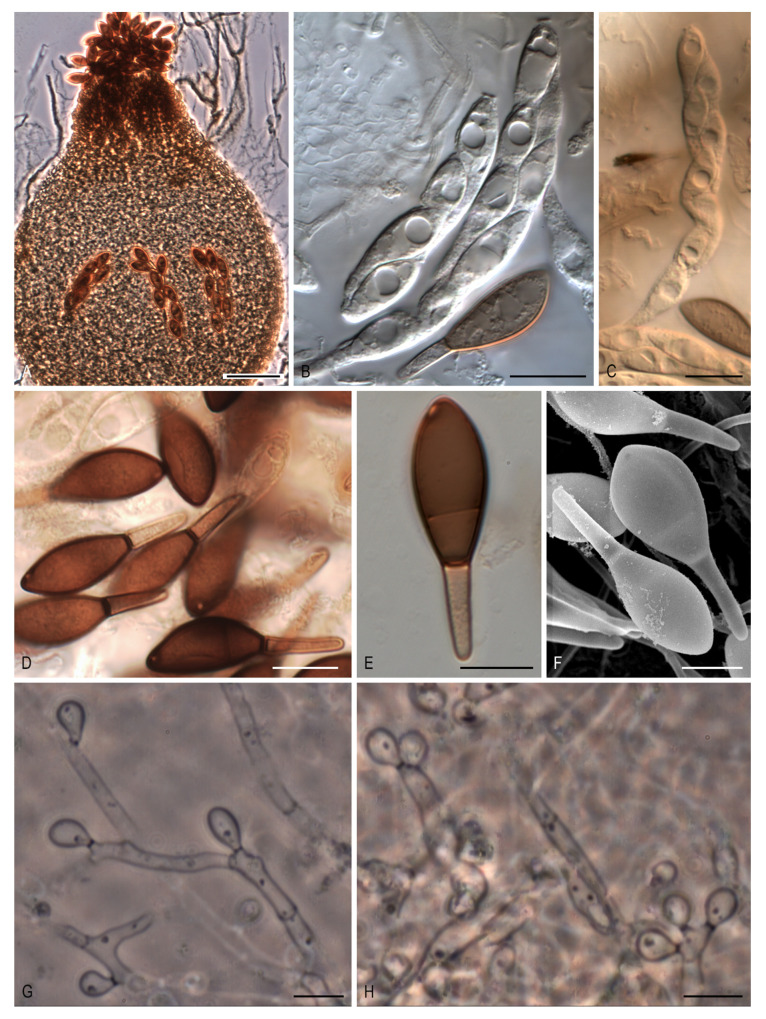
*Naviculispora terrestris* CBS 137295^T^. (**A**) Ascomata. (**B**,**C**) Asci. (**D**–**F**) Ascospores. (**G**) Conidia sessile. (**H**) Conidia borne on sympodially proliferating conidiophores. Scale bars: (**A**) = 100 μm; (**B**,**C**) = 20 μm; (**D**) = 15 μm; (**E**,**F**) = 10 μm; (**G**,**H**) = 5 μm.

**Figure 5 microorganisms-08-01430-f005:**
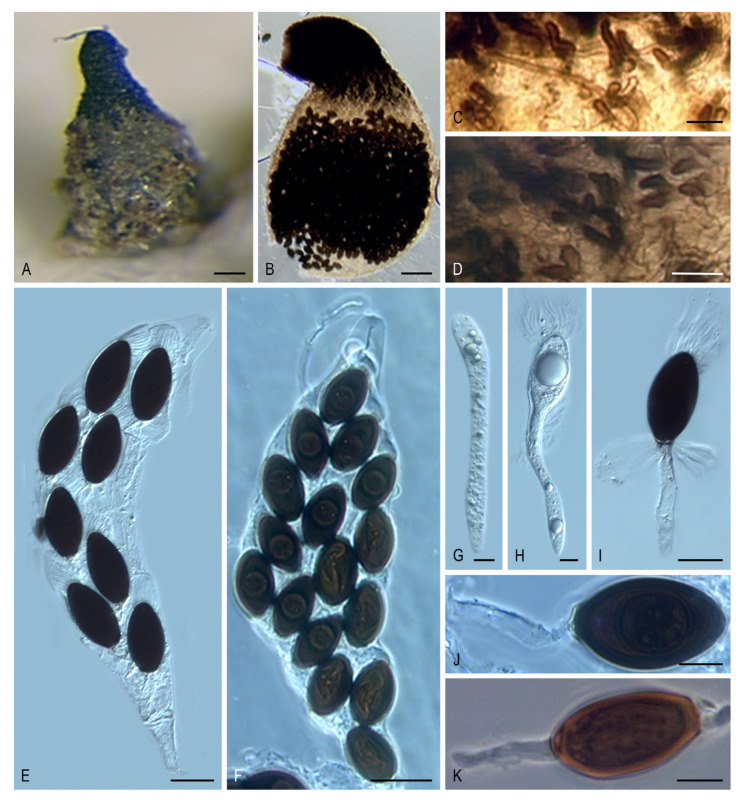
Morphology of *Rhypophila*. Ascoma. (**A**) *Rhypophila pleiospora* LyRS9223.1. (**B**) *Rhypophila myriospora* CBS 115804. Elongated tubercles at the neck. (**C**) *Rhypophila pleiospora* LyRS9223.1. (**D**) *Rhypophila myriospora* CBS 115804. Asci. (**E**) *Rhypophila decipiens* LyRS8109.2. (**F**) *Rhypophila pleiospora* LyRS9223.1. Immatured ascospores. (**G**,**H**) *Rhypophila decipiens* LyRS8109.2. Different ascospore morphologies. (**I**) *Rhypophila decipiens* LyRS8109.2. (**J**) *Rhypophila pleiospora* LyRS9223.1. (**K**) *Rhypophila myriospora* CBS 115804. Bars: (**A**,**B**) = 100 μm; (**C**,**D**) = 15 μm; (**E,F**) = 25 μm; (**G**,**H**,**J**,**K**) = 10 μm; (**I**) = 20 μm.

**Figure 6 microorganisms-08-01430-f006:**
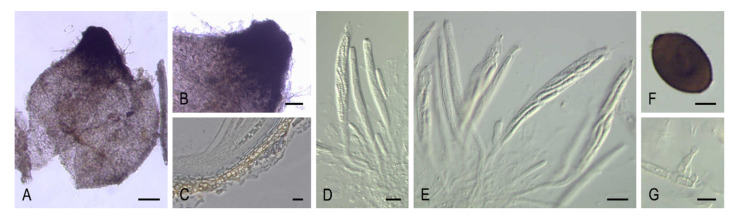
*Cladorrhinum coprophilum* SMH 3792. (**A**) Ascoma. (**B**) Neck. (**D,E**) Asci. (**F**) Ascospore. (**G**) Phialidic conidiogenous cell and conidia. *Cladorrhinum coprophilum* SMH 3083. (**C**) Ascomatal wall. Scale bars: (**A**) = 100 μm; (**B**) = 50 μm; (**C**–**E**) = 20 μm; (**F**,**G**) = 5 μm.

**Figure 7 microorganisms-08-01430-f007:**
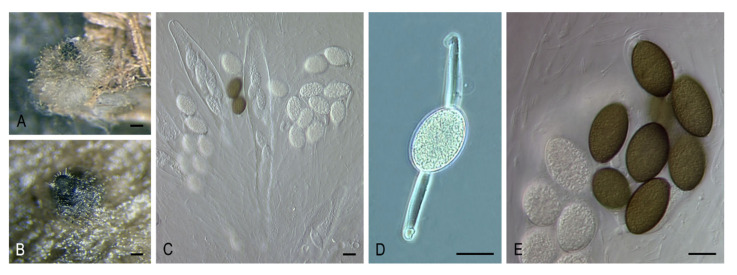
*Cladorrhinum foecundissimum* SMH 3253.3. (**A,B**) Ascomata. (**C**) Asci. (**D,E**) Ascospores. Bars: (**A,B**) = 200 μm; (**C**–**E**) = 20 μm. Pictures (**A,C**–**E**) adapted from [26].

**Figure 8 microorganisms-08-01430-f008:**
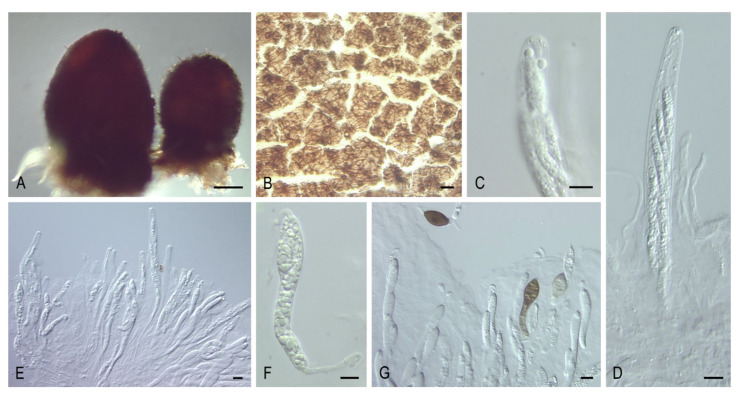
*Podospora costaricensis* GC 73. (**A**) Ascomata. (**B**) Ascomatal wall. (**C**) Ascal apex. (**D**,**E**) Asci. (**F**,**G**) Ascospores. Bars: (**A**) = 100 μm; (**B**–**E**, **G**) = 20 μm; (**F**) = 5 μm.

**Figure 9 microorganisms-08-01430-f009:**
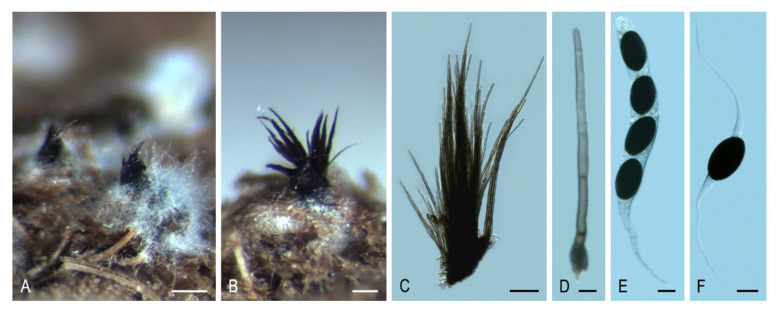
*Triangularia arizonensis* SMH 9134.1. (**A**,**B**) Ascomata. (**C**) Tuft of agglutinated hairs from the neck. (**D**) Hair from the neck. (**E**) Ascus and ascospores. (**F**) Ascospores. Bars: (**A**) = 200 μm; (**B**) = 100 μm; (**C**) = 50 μm; (**D**) = 5 μm; (**E**,**F**) = 20 μm.

**Figure 10 microorganisms-08-01430-f010:**
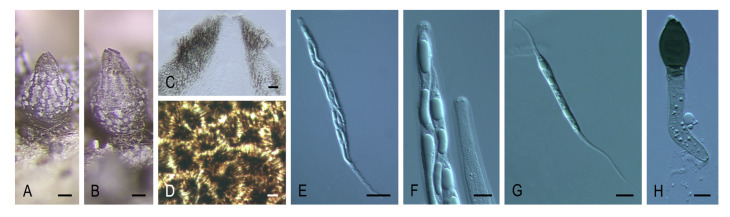
*Triangularia striata* SMH 3431. (**A**,**B**) Ascomata. (**C**) Neck. *Triangularia striata* SMH 4036. (**D**) Ascomal wall. (**E**) Ascus. (**F**) Ascus apex. (**G**,**H**) Ascospores. Bars: (**A**,**B**) = 100 μm; (**C**,**D**,**F**–**H**) = 10 μm; (**E**) = 50 μm. Pictures (**A**–**D**,**F**,**G**) adapted from [48].

**Figure 11 microorganisms-08-01430-f011:**
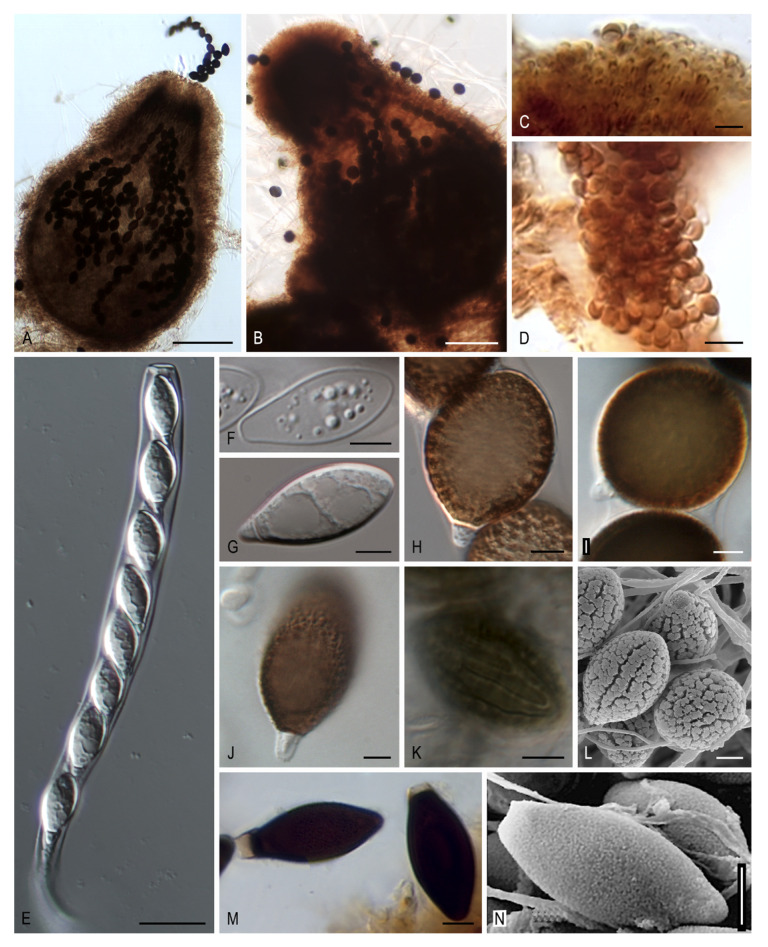
Morphology of *Jugulospora*. Ascoma. (**A**) *Jugulospora rotula* FMR 12428. (**B**) *Jugulospora rotula* FMR 12690. Detail of the papillate neck. (**C**,**D**) *Jugulospora carbonaria* ATCC 34657. Ascus. (**E**) *Jugulospora rotula* FMR 12428. Immature ascospores. (**F**,**G**) *Jugulospora rotula* FMR 12428. Different ascospore morphology of *Jugulospora rotula*. (**H**) CBS 12690. (**I**) CBS 110113. (**J**) CBS 110112. (**K**,**L**) FMR 12428. Ascospores. (**M**) *Jugulospora carbonaria* ATCC 34567. (**N**) *Jugulospora antarctica* IMI 381338. Bars: (**A**,**B**) = 100 μm; (**C**,**E**) = 20 μm; (**D**) = 10 μm; (**E**) = 15 μm; (**F**–**N**) = 5 μm.

**Figure 12 microorganisms-08-01430-f012:**
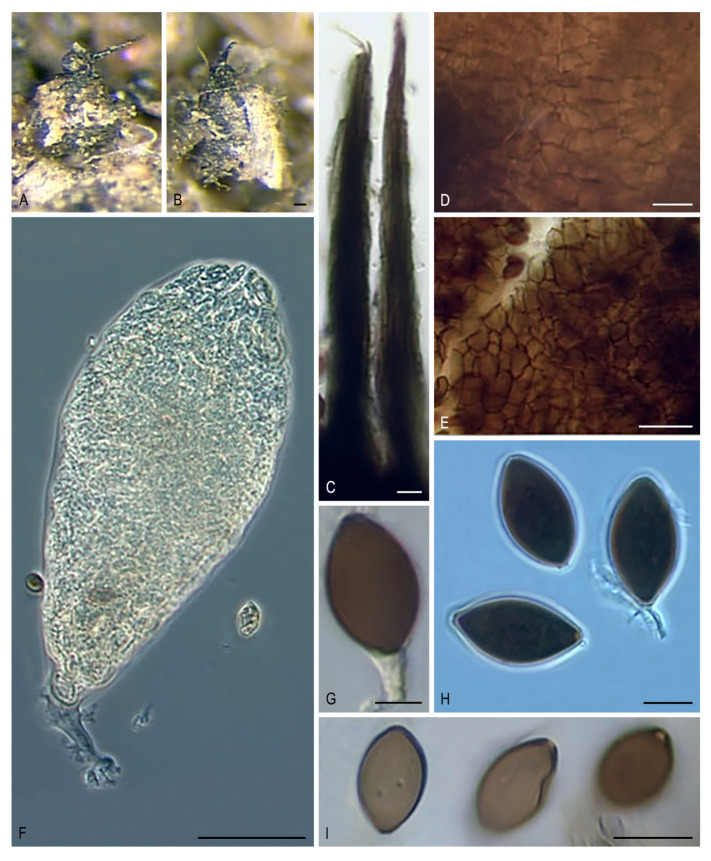
Morphology of *Pseudoechria.* Ascoma. (**A**,**B**) *Pseudoechria curvicolla* SMH 4381. Tuft of rigid hair of the neck. (**C**) *Pseudoechria curvicolla* SMH 4381. Ascomatal wall. (**D**). *Pseudoechria decidua* CBS 254.71^T^. (**E**). *Pseudoechria prolifica* CBS 250.71^T^. Ascus. (**F**) *Pseudoechria curvicolla* SMH 4381. Ascospores. (**G**) *Pseudoechria curvicolla* SMH 4381. (**H**) *Pseudoechria decidua* CBS 254.71^T^. (**I**). *Pseudoechria prolifica* CBS 250.71^T^. Bars: (**A**,**B**) = 100 μm; (**C**) = 20 μm; (**D**,**E**,**H**,**I**) = 10 μm; (**F**) = 50 μm; (**G**) = 5 μm.

**Figure 13 microorganisms-08-01430-f013:**
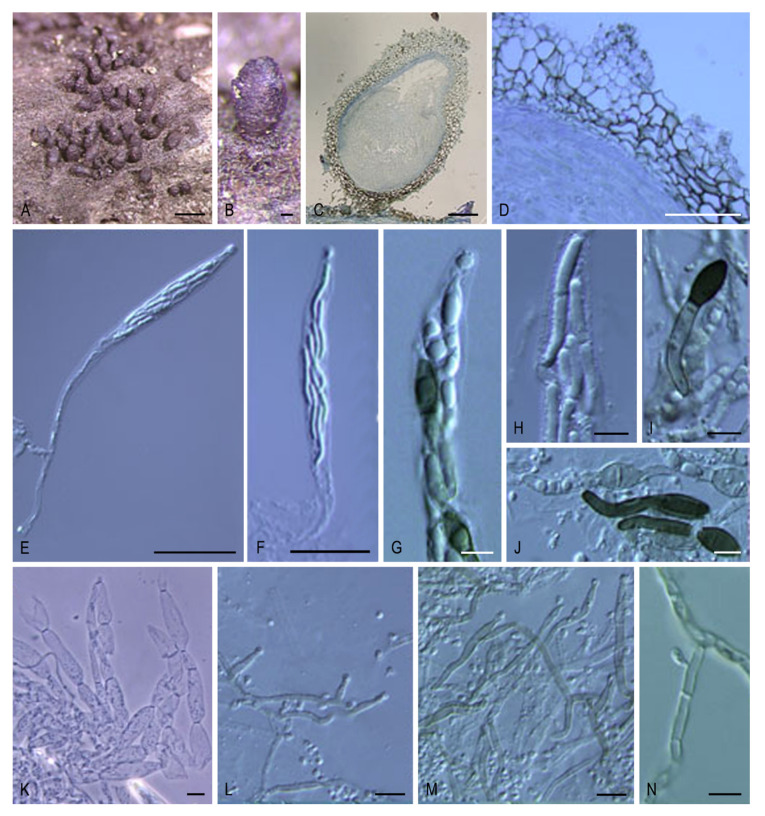
*Pseudoschizothecium atropurpureum* SMH 3073. (**A**–**C**) Ascomata. (**D**) Ascomatal wall. (**E**,**F**) Asci. (**G**,**H**) Ascus apex. (**I**,**J**) Ascospores. (**K**) Paraphyses. (**L**–**N)** Conidiogenous cells and conidia. Bars: (**A**) = 1 mm; (**B**,**C**) = 100 μm; (**D**–**F**) = 50 μm; (**G**–**N**) = 10 μm. All pictures adapted from [48].

**Table 1 microorganisms-08-01430-t001:** Isolates and reference strains of the order Sordariales included in this study. # GenBank accession numbers in bold were newly generated in this study. Taxonomic novelties are indicated in ***bold italic***.

Taxa	Strain	Source	GenBank Accession #			
			LSU	ITS	*rpb2*	*tub2*
*Anopodium ampullaceum **	MJR 40/07	GenBank, Kruys et al. [26]	KF557662	-	-	KF557701
	E00218015	GenBank, Kruys et al. [26]	KF557663	-	-	KF557702
*Apiosordaria microcarpa **	CBS 692.82^T^	GenBank, Wang et al. [6]	MK926841	MK926841	MK876803	-
*Apodospora gotlandica*	E00204952	GenBank, Kruys et al. [26]	KF557664	-	-	KF557703
*Apodospora simulans*	Kruys 701	GenBank, Kruys et al. [26]	KF557666	-	-	KF557704
	n/a	GenBank, Kruys et al. [26]	KF557667	-	-	KF557705
***Areotheca ambigua***	CBS 215.60	GenBank, Cai et al. [4]	AY999114	AY999137	-	-
***Areotheca areolata***	UAMH 7495	GenBank, Miller and Huhndorf [27]	AY587936	AY587911	AY600275	AY600252
*Arnium caballinum **	Lundqvist 7098-e	GenBank, Kruys et al. [26]	KF557672	-	-	-
*Arnium cirriferum **	CBS 120041	GenBank, Kruys et al. [26]	KF557673	-	-	KF557709
*Arnium hirtum **	E00204950	GenBank, Kruys et al. [26]	KF557675	-	-	KF557711
	E00204487	GenBank, Kruys et al. [26]	KF557676	-	-	KF557712
*Arnium japonense **	SANK 10273	GenBank, Kruys et al. [26]	KF557680	-	-	KF557713
*Arnium mendax **	Lundqvist 20874-c	GenBank, Kruys et al. [26]	KF557687	-	-	KF557716
	E00122117	GenBank, Kruys et al. [26]	KF557688	-	-	KF557717
*Bellojisia rhynchostoma **	CBS 118484	GenBank, Réblová [28]	EU999217	-	-	-
*Bombardia bombarda*	AR1903	GenBank, Miller and Huhndorf [11]	AY780052	-	AY780152	AY780089
	SMH 3391	GenBank, Huhndorf et al. [2], Miller and Huhndorf [11]	AY346263	-	AY780153	AY780090
	SMH 4821	GenBank, Miller and Huhndorf [11]	AY780053	-	AY780154	AY780091
*Bombardioidea anartia **	HHB99-1	GenBank, Huhndorf et al. [2], Miller and Huhndorf [11]	AY346264	-	AY780155	AY780092
*Camarops amorpha*	SMH 1450	GenBank, Miller and Huhndorf [11]	AY780054	-	AY780156	AY780093
*Cercophora mirabilis*	CBS 120402	Wallaby dung, Australia, Victoria, Eucalyptus forest near Healesville	**KP981429**	**MT784128**	**KP981611**	**KP981556**
*Cercophora scortea **	GJS L556	GenBank, Miller and Huhndorf [11]	AY780063	-	AY780168	AY780107
*Cercophora sparsa **	JF 00229	GenBank, Miller and Huhndorf [27]	AY587937	AY587912	-	AY600253
*Cercophora sulphurella **	SMH 2531	GenBank, Miller and Huhndorf [27]	AY587938	AY587913	AY600276	AY600254
*Chaetomium globosum*	CBS 160.62^T^	GenBank, Vu et al. [29], Wang et al. [30]	MH869713	KT214565	KT214666	-
***Cladorrhinum coprophilum***	SMH 3794	GenBank, Miller and Huhndorf [11]	AY780058	-	AY780162	AY780102
*Cladorrhinum foecundissimum*	CBS 180.66^T^	GenBank, Wang et al. [6]	MK926856	MK926856	MK876818	-
	CBS 120012	GenBank, Kruys et al. [26]	KF557689	-	-	KF557718
*Cladorrhinum hyalocarpum*	CBS 322.70^T^	GenBank, Wang et al. [6]	MK926857	MK926857	MK876819	-
*Cladorrhinum hyalocarpum*	FMR 13412	Soil, India, Gualior	**KP981428**	**MT784129**	**KP981610**	**KP981555**
*Cladorrhinum intermedium*	CBS 433.96^T^	GenBank, Wang et al. [6]	MK926859	MK926859	MK876821	-
***Cladorrhinum tomentosum***	Francoise Candoussau	GenBank, Kruys et al. [26]	KF557691	-	-	KF557720
*Corylomyces selenosporus **	CBS 113930^T^	Dry fruti, France, Saint Pé de Bigorre; GenBank, Stchigel et al. [31]	DQ327607	**MT784130**	**KP981612**	**KP981557**
*Corynascus sepedonium*	CBS 111.69^T^	GenBank, Vu et al. [29], Greif et al. [32]	MH871003	MH859271	FJ666394	-
*Corynascella humicola*	CBS 337.72^T^	GenBank, Vu et al. [29]	MH872209	MH860493	-	-
*Diplogelasinospora grovesii*	CBS 340.73^T^	GenBank, Vu et al. [29]	MH872401	MH860693	-	-
*Diplogelasinospora inaequalis*	CBS 436.74^T^	GenBank, Cai et al. [33]	AY681167	AY681201	-	-
*Diplogelasinospora moalensis*	CBS 136018^T^	Soil, Spain, Principado de Asturia, Cangas del Narcea, Veiga de Rengos, Moal; Genbank, Crous et al. [34]	**KP981430**	HG514152	**KP981613**	**KP981558**
*Diplogelasinospora princeps*	FMR 13414	Soil, Tennessee, Great Smoky Mountains National Park	**KP981431**	**MT784131**	**KP981614**	**KP981559**
*Diplogelasinospora princeps*	FMR 13415	Soil, Tennessee, Great Smoky Mountains National Park	**KP981432**	**-**	**KP981615**	**KP981560**
*Echria gigantospora*	F77-1	GenBank, Kruys et al. [26]	KF557674	-	-	KF557710
*Echria macrotheca*	Lundqvist 2311	GenBank, Kruys et al. [26]	KF557684	-	-	KF557715
*Fimetariella rabenhorstii*	Lundqvist 20410-c	GenBank, Kruys et al. [26]	KF557694	-	-	KF557721
*Immersiella caudata*	SMH 3298	GenBank, Miller and Huhndorf [10,11]	AY436407	-	AY780161	AY780101
*Immersiella immersa*	SMH 4104	GenBank, Miller and Huhndorf [10,11]	AY436409	-	AY780181	AY780123
***Jugulospora antarctica***	IMI 381338^T^	Soil, Antarctica, King George Island, Jubany Argentinian base	**KP981433**	-	**KP981616**	**KP981561**
***Jugulospora carbonaria***	ATCC 34567	GenBank, Huhndorf et al. [2], Miller and Huhndorf [11]	AY346302	-	AY780196	AY780141
*Jugulospora rotula*	ATCC 38359	GenBank, Huhndorf et al. [2], Miller and Huhndorf [11]	AY346287	-	AY780178	AY780120
	CBS 110112	Soil, Spain, Tarragona, Gorgs de la Febró	**KP981434**	**-**	**KP981617**	**KP981562**
	CBS 110113	Soil, Spain, Tarragona, Gorgs de la Febró	**KP981435**	**-**	**KP981618**	**KP981563**
	FMR 12428	Soil, Tennessee, Great Smoky Mountains National Park	**KP981436**	**MT784132**	**KP981619**	**KP981564**
	FMR 12690	Soil, Tennessee, Great Smoky Mountains National Park	**KP981437**	**MT784133**	**KP981620**	**KP981565**
	FMR 12781	Soil, Tennessee, Great Smoky Mountains National Park	**KP981438**	**MT784134**	**KP981621**	**KP981566**
***Jugulospora vestita***	CBS 135.91^T^	Soil, Neche, Nepal	**MT785872**	**MT784135**	**MT783824**	**MT783825**
*Lasiosphaeria glabrata*	TL 4529	GenBank, Miller and Huhndorf [10,27]	AY436410	AY587914	AY600277	AY600255
*Lasiosphaeria lanuginosa*	SMH 3819	GenBank, Miller and Huhndorf [10,27]	AY436412	AY587921	AY600283	AY600262
*Lasiosphaeria ovina*	SMH 1538	GenBank, Miller and Huhndorf [27], Fernandez et al. [35,36]	AF064643	AY587926	AY600287	AF466046
*Lasiosphaeria rugulosa*	SMH 1518	GenBank, Miller and Huhndorf [10,27]	AY436414	AY587933	AY600294	AY600272
*Lasiosphaeria sorbina*	CBS 885.85	GenBank, Miller and Huhndorf [10,27]	AY436416	AY587935	AY600296	AY600274
*Lasiosphaeris hirsuta*	SMH 1543	GenBank, Miller and Huhndorf [10,11]	AY436417	-	AY780179	AY780121
*Lasiosphaeris hispida*	SMH 3336	GenBank, Miller and Huhndorf [10,11]	AY436419	-	AY780180	AY780122
***Lundqvistomyces karachiensis***	CBS 657.74	Arid soil, Egypt, Western Desert, Kharga Oasis; GenBank, Wang et al. [6]	**KP981447**	MK926850	**KP981630**	**KP981478**
***Lundqvistomyces tanzaniensis***	TRTC 51981^T^	GenBank, Miller and Huhndorf [11], Vu et al. [29]	AY780081	MH862260	AY780197	AY780143
***Naviculispora terrestris***	CBS 137295^T^	Soil, Argentina, Tucumán province, Tafí del Valle	**KP981439**	**MT784136**	**KP981622**	**KP981567**
*Neurospora pannoica*	TRTC 51327	GenBank, Miller and Huhndorf [11]	AY780070	-	AY780185	AY780126
*Podospora appendiculata **	CBS 212.97	GenBank, Miller and Huhndorf [11], Vu et al. [29]	AY780071	MH862644	AY780186	AY780129
*Podospora bulbillosa*	CBS 304.90^T^	GenBank, Wang et al. [6]	MK926861	MK926861	MK876823	-
***Podospora costaricensis***	SMH 4021	GenBank, Miller and Huhndorf [11]	AY780059	-	AY780163	AY780103
*Podospora didyma **	CBS 232.78	GenBank, Cai et al. [4]	AY999100	AY999127	-	-
*Podospora fabiformis **	CBS 112043^T^	GenBank, Wang et al. [6]	MK926843	MK926843	MK876805	-
*Podospora fibrinocaudata **	CBS 315.91^T^	GenBank, Wang et al. [6]	MK926844	MK926844	MK876806	-
	TRTC 48343	GenBank, Miller and Huhndorf [11]	AY780074	-	AY780188	AY780131
*Podospora fimicola*	CBS 482.64^ET^	Dung of cow, Switzerland, Kt. Aargau, Ober-Erlinsbach, Barmelweid; GenBank, Wang et al. [6]	**KP981440**	MK926862	**KP981623**	**KP981568**
***Podospora sacchari***	CBS 713.70^T^	Root, Jamaica, Janswood Estates; GenBank, Vu et al. [29]	**KP981425**	MH859915	**KP981607**	**KP981552**
***Podospora striatispora***	CBS 154.77^T^	Soil, Thailand, Sukhotai	**KP981426**	**MT784137**	**KP981608**	**KP981553**
***Pseudoechria curvicolla***	IFO 8548	GenBank, Cai et al. [4]	AY999099	AY999122	-	-
	CBS 259.69	GenBank, Vu et al. [29]	MH871036	MH859302	-	-
***Pseudoechria decidua***	CBS 254.71^T^	GenBank, Wang et al. [6]	MK926842	MK926842	MK876804	-
***Pseudoechria longicollis***	CBS 368.52^T^	GenBank, Wang et al. [6]	MK926847	MK926847	MK876809	-
***Pseudoechria prolifica***	CBS 250.71^T^	GenBank, Wang et al. [6]	MK926848	MK926848	MK876810	-
*Pseudoneurospora canariensis*	FMR 12156^T^	GenBank, Vu et al. [29], Crous et al. [34]	MH877580	-	-	HG423208
***Pseudoschizothecium atropurpureum***	SMH 2961	GenBank, Miller and Huhndorf [11]	AY780056	-	-	AY780099
	SMH 3073	GenBank, Miller and Huhndorf [11]	AY780057	-	AY780160	AY780100
*Rinaldiella pentagonospora*	CBS 132344^T^	Contaminated human lesion, USA, Georgia, Dahlonega, GenBank, Vu et al. [29]	**KP981442**	MH866007	**KP981625**	**KP981570**
***Rhypophila cochleariformis***	CBS 249.71	GenBank, Cai et al. [4]	AY999098	AY999123	-	-
***Rhypophila decipiens***	CBS 258.69	GenBank, Miller and Huhndorf [11], Miller [unpubl. data]	AY780073	KX171946	AY780187	AY780130
***Rhypophila myriaspora***	TNM F17211	GenBank, Chang et al. [12]	-	EF197083	-	-
***Rhypophila pleiospora***	TNM F16889	GenBank, Chang et al. [12]	-	EF197084	-	-
*Schizothecium curvisporum*	ATCC 36709	GenBank, Huhndorf et al. [2], Miller and Huhndorf [11]	AY346300	-	AY780192	AY780136
*Schizothecium fimbriatum*	CBS 144.54	GenBank, Cai et al. [4], Miller and Huhndorf [11]	AY780075	AY999115	AY780189	AY780132
*Schizothecium inaequale*	CBS 356.49^T^	GenBank, Wang et al. [6]	MK926846	MK926846	MK876808	-
***Schizothecium selenosporum***	CBS 109403^T^	GenBank, Wang et al. [6]	MK926849	MK926849	MK876811	-
*Sordaria fimicola*	SMH 4106	GenBank, Miller and Huhndorf [11]	AY780079	-	AY780194	AY780138
*Triangularia allahabadensis*	CBS 724.68^T^	GenBank, Wang et al. [6]	MK926865	MK926865	MK876827	-
*Triangularia anserina*	CBS 433.50	GenBank, Wang et al. [6]	MK926864	MK926864	MK876826	-
***Triangularia arizonensis***	Santesson 18211-c	GenBank, Kruys et al. [26]	KF557668	-	-	KF557706
	Kruys 724	GenBank, Kruys et al. [26]	KF557669	-	-	KF557707
	E00204509	GenBank, Kruys et al. [26]	KF557670	-	-	KF557708
	CBS 120289	GenBank, Debuchy et al. [unpubl. data]	KU955584	-	-	-
*Triangularia backusii*	CBS 539.89^IsoT^	GenBank, Wang et al. [6]	MK926866	MK926866	MK876828	-
*Triangularia backusii*	FMR 12439	Soil, Tennessee, Great Smoky Mountains National Park	**KP981423**	**MT784138**	**KP981605**	**KP981550**
*Triangularia backusii*	FMR 13591	Soil, Spain, Tarragona, Els Gorgs de la Febró	**KP981424**	**MT784139**	**KP981606**	**KP981551**
*Triangularia bambusae*	CBS 352.33^T^	GenBank, Wang et al. [6]	MK926868	MK926868	MK876830	-
*Triangularia batistae*	CBS 381.68^T^	Soil, Brazil	**KP981443**	**MT784140**	**KP981626**	**KP981577**
*Triangularia longicaudata*	CBS 252.57^T^	GenBank, Wang et al. [6]	MK926871	MK926871	MK876833	-
*Triangularia longicaudata*	FMR 12365	Soil, Tennessee, Great Smoky Mountains National Park	**KP981448**	**MT784141**	**KP981631**	**KP981474**
*Triangularia longicaudata*	FMR 12782	Soil, Spain, Gran Canaria	**KP981449**	**MT784142**	**KP981632**	**KP981475**
*Triangularia mangenotii **	CBS 419.67^T^	Leaf, France, near Bordeaux	**KP981444**	**MT784143**	**KP981627**	**KP981571**
*Triangularia pauciseta*	CBS 451.62	GenBank, Wang et al. [6]	MK926870	MK926870	MK876832	-
*Triangularia phialophoroides*	CBS 301.90^T^	GenBank, Wang et al. [6]	MK926871	MK926871	MK876833	-
*Triangularia setosa*	FMR 12787	Spain, Gran Canaria	**KP981441**	**MT784144**	**KP981624**	**KP981569**
	CBS 311.58	GenBank, Wang et al. [6]	MK926872	MK926872	MK876834	-
***Triangularia striata***	SMH 3431	GenBank, Miller and Huhndorf [11]	-	AY780065	AY780169	AY780108
	SMH 4036	GenBank, Miller and Huhndorf [11], Miller [unpubl. data]	KX348038	AY780066	-	-
***Triangularia tetraspora***	CBS 245.71	GenBank, Vu et al. [29]	MH860097	MH871873	-	-
	FMR 5770	GenBank, Cai et al. [unpubl. data]	AY999130	AY999108	-	-
*Triangularia verruculosa*	CBS 148.77	GenBank, Wang et al. [6]	MK926874	MK926874	MK876836	-
*Zopfiella attenuata **	CBS 266.77^T^	Soil, Japan; GenBank, Vu et al. [29]	**KP981445**	MH861060	**KP981628**	**KP981572**
*Zopfiella marina **	CBS 155.77^T^	GenBank, Wang et al. [6]	MK926851	MK926851	MK876813	-
*Zopfiella pleuropora **	CBS 518.70^T^	Dung of deer, Ontario, Haliburton Co., S of Dorset	**KP981450**	**MT784145**	**KP981633**	**KP981476**
*Zopfiella pilifera **	CBS 413.73^T^	GenBank, Wang et al. [6]	MK926852	MK926852	MK876814	-
*Zopfiella submersa **	CBS 698.96^T^	GenBank, Wang et al. [6]	MK926853	MK926853	MK876815	-
*Zopfiella tabulata*	CBS 230.78	GenBank, Wang et al. [6]	MK926854	MK926854	MK876816	-
*Zopfiella tardifaciens **	CBS 670.82^T^	GenBank, Wang et al. [6]	MK926855	MK926855	MK876817	-
*Zygopleurage zygospora*	SMH 4219	GenBank, Huhndorf et al. [2], Miller and Huhndorf [11]	AY346306	-	-	AY780147
*Zygospermella insignis*	Lundqvist 2444	GenBank, Kruys et al. [26]	KF557698	-	-	KF557722
	E00204312	GenBank, Kruys et al. [26]	KF557699	-	-	KF557723

ATCC: American Type Culture Collection, Virginia, USA; CBS: Westerdijk Fungal Biodiversity Institute, Utrecht, the Netherlands; FMR: Facultat de Medicina, Reus, Spain; IFO: Biological Resource Center, Chiba, Japan; IMI: International Mycological Institute, CABI-Bioscience, Egham, UK; SANK: Research laboratories of the Daiichi Sanko Pharmaceutical Co., Ltd., Tokyo, Japan; TNM: Herbarium of National Museum of Natural Science, Taiwan; TRTC: Royal Ontario Museum, Toronto, Canada; UAMH: UAMH Center for Global Microfungal Biodiversity, University of Toronto, Canada; AR, Francoise Candoussau, GJS, JF, HHB, Kruys, Lundqvist, MJR, Santensoon, SMH, TL: Personal collections of Amy Rossman, Francoise Candoussau, Gary J. Samuels, Jacques Fournier, Harold H. Burdsal, Åsa Kruys, Nils Lundqvist, Michael J. Richardson, Sweden R. Santesson, Sabine M. Huhndorf, Thomas Læssøe, respectively; n/a: Not available. ^ET, IsoT and T^ indicates ex-epitype, ex-isotype, and ex-type strains, respectively. * Taxa with generic names applied in the broad sense (sensu lato), not necessarily reflecting molecular phylogenetic relationships.

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
