# Peer review of "Re-Evaluation of the Order Sordariales: Delimitation of Lasiosphaeriaceae s. str., and Introduction of the New Families Diplogelasinosporaceae, Naviculisporaceae, and Schizotheciaceae"

_microorganisms, 2020, doi:10.3390/microorganisms8091430_

Round 1

Reviewer 1 Report

Dear Authors, 

please check annotated comments.

Author Response

Comments are answered in the pdf attached, as well as here:

Comments Reviewer 1.

  1. (line 87) Please insert the degrees symbol. check throughout the text.

This change was made throughout the text.

  1. (line 93) cultured; throughout the text

Changed.

  1. (line117) I wouldn't dare to say that without arguing that there is an "issue" with low support when combining in a matrix low-evolving and fast-evolving regions of DNA.

further required reading:

10.1111/j.1600-0587.2012.07773.x

10.1214/ss/1063994980

10.1007/s00606-017-1459-y

Nicholas D. Pattengale, Masoud Alipour, Olaf R.P. Bininda-Emonds,

Bernard M.E. Moret and Alexandros Stamatakis. How Many Bootstrap Replicates Are Necessary? S. Batzoglou (Ed.): RECOMB 2009, LNCS 5541, pp. 184–200, 2009

We agree there can be problems when bootstrapping is applied to sequences that vary in their rate of evolution, but we decided to employ the most common support values considered significant in fungal taxonomy, which are widely used and effectiveness corroborated by Alfaro et al. [26].

  1. (page 10) Underlined several times Cercophora and Podospora

If the reviewer wants to point out that species of Cercophora and Podospora are scattered throughout the order Sordariales, we completely agree with the observation, which demonstrates that the current taxonomic classification of lasiosphaeriaceous taxa is so far artificial. In fact, Podospora has been recently re-delimited by Wang et al. [6]. However, a high number of species of both genera are still waiting to be taxonomically re-classified. The proper delimitation of Cercophora could not be performed in the present study because of the lack of sequences of the type strain of Cercophora mirabilis, which is the type species of the genus, and also the lack of sporulation of the strain of this species included in the study (being not possible the typification), which was explained in Results (line 727-730) and Discussion (1127-1134) chapters.

  1. (line 165) (i.e. ......)

Changed.

  1. (line 267) the phylogentic inference resulted in both several sister clades of Arnium inside Naviculisporaceae and Schizotheciaceae.

Please explain both your proposal of new families and the addressing genera which are spread even beyond the proposed families (for instance Zopfiella)

The problem about delimitation of Arnium and Zopfiella, as well as Cercophora, was already explained in detail in the discussion (lines 1119-1165). The presence of species of the same genus in different families does not indicate that the families are not correctly proposed, but it demonstrates that the traditional taxonomic classification based on ascospore morphology is artificial. All these arguments are developed in the Discussion (lines 1069-1081). We have considered that to explain this problem separately for each family introduced would have made the results repetitive and unnecessarily long.

The proposal of the new families that is above the description of each family is explained based on the phylogenetic inferences (lines 163-167, 237-240, 680-683).

  1. (line 486) Please keep throughout the text only the first letter of the genus

To keep only the first letter for the abbreviation of the genera makes the reading of the manuscript difficult and confusing, since Apiosordaria and Arnium, as well as Cladorrhinum and Cercophora, Podospora, Pseudoechria and Pseudoschizothecium, and Triangularia and Thielavia would have the same abbreviation. The use of more than one letter for genus abbreviation is actually rather common (e.g., doi: 10.1016/j.simyco.2016.11.005.)

  1. (line 542) not quite...only P. striatispora

The term branches was substituted by lineages.

  1. (line 680) cannot see/understand the argument of being a strong supported clade...

for a new family it should be i) more type seqs, ii) explanations on other species outside the clade; iii) morphological parameters relevant for systematics discussed, iv) more existent studies to correlate the information

  1. i) To date, our work is the one that includes sequence data of most ex-type strains. Regrettably, type material for a large number of lasiosphaeriaceous species are not available, and a future goal is to recollect material to epitypify them and to validate their names. However, we do not consider that this difficulty should lead to continued use of an artificial taxonomic classification of the polyphyletic lasiosphaeriaceous taxa until more material will be collected, because it could take several years to complete this goal.
  2. ii) The explanation about other species outside the clades (we think the reviewer means those species of the same genus in different clades) are given in the Discussion (lines 1096-1165). It is widely demonstrated that the taxonomic placement of most lasiosphaeriaceous genera are artificial since 2004 [10]. For that reason, the presence of species of Apiosordaria, Podospora and Triangularia in other clades did not limit the introduction of the family Podosporaceae in 2019 [6], as well as we do not consider it needs to block the delimitation of monophyletic families to arrive to a correct classification of the lasiosphaeriaceous taxa.

iii) Morphological parameters of each family introduced are extensively discussed in the Discussion (lines 1035-1068). While the morphological characteristics to distinguish Diplogelasinosporaceae are evident, these are not that clear between the other families, including the already established families Podosporaceae and Chaetomiaceae. The problem in the correlation between morphology and phylogeny is also widely explained in the Discussion (lines1069-1081).

  1. iv) Our proposals are in concordance with those of previous studies since the separation of Lasiosphaeriaceae sensu lato in four different clades was already noted by Kruys et al. in 2015 [13], and nowadays we are working on the proper delimitation of these monophyletic clades, such as Wang et al. (2019) [6] and in our study.

  1. (line 681) one species (highlighting Rinaldiella)

Yes, only one species since Rinaldiella is a monospecific genus.

  1. (line 681) several species of Arnium are clustered in Naviculisporaceae

Yes, it is explained in the Discussion to avoid repetition along the Results, as was mentioned in all the comments before.

  1. (line 681) Lasiosphaeriaceae... (highlighting Cercophora)

We suppose that the reviewer wants to point out that species of Cercophora are present in other clades before having been considered as Lasiosphaeriaceae sensu lato. Yes, several species of Cercophora and Podospora are scattered throughout different families of the Sordariales, as we have explained in a previous answer (4). This fact is due to the artificial circumscription of the lasiosphaeriaceous genera. Podospora has been recently delimited by Wang et al. [6]. However, lots of Podospora spp. are still waiting for a correct placement. On the other hand, the delimitation of Cercophora could not be performed in the present study because of the lack of sequences from the type strain of Cercophora mirabilis, the type species of the genus. Further studies are necessary to solve this problem. This issue is discussed in the Results and Discussion (727-730, 1127-1134) of the manuscript.

Reviewer 2 Report

The manuscript "Re-evaluation of the order Sordariales: delimitation of Lasiosphaeriaceae and introduction of the new families Diplogelasinosporaceae, Naviculisporaceae, and Schizothecieae" by Marin-Felix et al describes a reclassification of the polyphyletic Lasiosphaeriaceae family into four families with six new genera based on morphological and sequencing data.  The authors clearly performed an exhaustive molecular and morphological characterization of the Lasiosphaeriaceae family and the conclusions of the authors are justified based on the results.  No changes are suggested.

Author Response

There is no changes suggested from the reviewer 2.